# A *Drosophila* screen identifies NKCC1 as a modifier of NGLY1 deficiency

Dana M Talsness[1†], Katie G Owings[1†], Emily Coelho[1], Gaelle Mercenne[2], John M Pleinis[2], Raghavendran Partha[3], Kevin A Hope[1], Aamir R Zuberi[4], Nathan L Clark[1], Cathleen M Lutz[4], Aylin R Rodan[2,5], Clement Y Chow[1]*

[1]Department of Human Genetics, University of Utah School of Medicine, Salt Lake City, United States; [2]Department of Internal Medicine, Division of Nephrology and Hypertension, and Molecular Medicine Program, University of Utah, Salt Lake City, United States; [3]Department of Computational and Systems Biology, University of Pittsburgh, Pittsburgh, United States; [4]Genetic Resource Science, The Jackson Laboratory, Bar Harbor, United States; [5]Medical Service, Veterans Affairs Salt Lake City Health Care System, Salt Lake City, United States

**Abstract** N-Glycanase 1 (NGLY1) is a cytoplasmic deglycosylating enzyme. Loss-of-function mutations in the *NGLY1* gene cause NGLY1 deficiency, which is characterized by developmental delay, seizures, and a lack of sweat and tears. To model the phenotypic variability observed among patients, we crossed a *Drosophila* model of NGLY1 deficiency onto a panel of genetically diverse strains. The resulting progeny showed a phenotypic spectrum from 0 to 100% lethality. Association analysis on the lethality phenotype, as well as an evolutionary rate covariation analysis, generated lists of modifying genes, providing insight into NGLY1 function and disease. The top association hit was *Ncc69* (human *NKCC1/2*), a conserved ion transporter. Analyses in *NGLY1*-/- mouse cells demonstrated that NKCC1 has an altered average molecular weight and reduced function. The misregulation of this ion transporter may explain the observed defects in secretory epithelium function in NGLY1 deficiency patients.

*For correspondence:
cchow@genetics.utah.edu

†These authors contributed equally to this work

Competing interests: The authors declare that no competing interests exist.

## Introduction

NGLY1 deficiency (OMIM 615273) is a rare, autosomal recessive disorder caused by loss-of-function mutations in the *NGLY1* gene. Patients with NGLY1 deficiency have a variety of symptoms, including developmental delay, seizures, liver dysfunction, central and peripheral nervous system abnormalities, sweat gland abnormalities, and a lack of tears (alacrima) (*Enns et al., 2014*; *Lam et al., 2017*). While the first NGLY1 deficiency patient was only recently identified (*Need et al., 2012*), there have been rapid research advances thanks to the support of two patient organizations (NGLY1.org and Grace Science Foundation). Even though a great deal has been learned about the genetic disorder in a short amount of time, there are currently no cures or approved treatments for NGLY1 deficiency.

The *NGLY1* gene encodes the N-Glycanase protein (NGLY1). NGLY1 functions as part of the Endoplasmic Reticulum (ER) Associated Degradation (ERAD) pathway as evidenced by its association with other ERAD components (*Katiyar et al., 2005*; *McNeill et al., 2004*; *Park et al., 2001*). The ERAD pathway retrotranslocates misfolded proteins from the ER lumen to the cytoplasm where they are degraded by the proteasome (reviewed in *Qi et al., 2017*). NGLY1 is localized to the cytoplasm where it is thought to remove N-linked glycans from misfolded proteins prior to their degradation (*Hirsch et al., 2003*). Recent evidence suggests that this deglycosylation is required for retrotranslocation for at least some protein substrates (*Galeone et al., 2020*). Nevertheless, it remains unclear whether NGLY1 is required to deglycosylate all misfolded proteins, or just a subset, or if it is

necessary for protein degradation at all. It has been shown that model substrates can be degraded regardless of glycosylation state (*Hirsch et al., 2003*; *Kario et al., 2008*). While a recent report showed that ER stress markers were increased in NGLY1 -/- MEFs (*Galeone et al., 2020*), other experiments such as RNAi knockdown (KD) of NGLY1 in *Drosophila* (*Owings et al., 2018*) and loss of NGLY1 function in mouse, rat, and human cells (*Asahina et al., 2020*; *Mueller et al., 2020*; *Tambe et al., 2019*) have shown no evidence of ER stress. ER stress is often observed when there are mutations in proteins that are necessary for ERAD due to the accumulation of misfolded proteins in the ER. It may be that NGLY1 is not necessary for ERAD, or it is involved in a non-canonical ERAD function, or it may be deglycosylating cytoplasmic proteins for an entirely different purpose. These hypothesized functions are not mutually exclusive.

NGLY1 has been shown to deglycosylate various exogenous model substrates such as TCR-α (*Hirsch et al., 2003*) and RNaseB (*Kario et al., 2008*). To identify endogenous substrates several mass spectrometry experiments have been performed (*Fujihira et al., 2017*; *Hosomi et al., 2016*; *Maynard et al., 2020*; *Zolekar et al., 2018*). Yet, the first high-confidence substrate of NGLY1 deglycosylation, NRF1 (gene: *NFE2L1*), was discovered in a *Caenorhabditis elegans* genetic screen (*Lehrbach and Ruvkun, 2016*). NRF1 mediates a proteasome 'bounce-back' response. NRF1 is constitutively degraded by the proteasome through the ERAD pathway, until the proteasome is inhibited or overwhelmed by protein load. During this proteasome stress, NRF1 accumulates and is deglycosylated by NGLY1 (*Tomlin et al., 2017*). Rather than targeting the protein for degradation, the deglycosylation activates NRF1 by converting asparagine to aspartic acid residues (*Lehrbach et al., 2019*). NRF1 can then be imported into the nucleus to act as a transcription factor for proteasome subunits. The lack of NRF1 activation in NGLY1-deficient patients likely explains some of the disorder's symptoms such as motor dysfunction (*Kobayashi et al., 2011*) and cognitive deficits (*Lee et al., 2011*). Recently, it was found that the protein BMP4 is deglycosylated by NGLY1 when overexpressed in either *Drosophila* or mammalian cells (*Galeone et al., 2020*). BMP4 is a signaling molecule and could explain several of the developmental symptoms of NGLY1 deficiency. However, not all symptoms can be explained by these two targets and therefore there is a pressing need to identify and validate more substrates of NGLY1 deglycosylation.

In addition to discovering new NGLY1 targets, there is a need to understand how background genetic variants affect the number and severity of symptoms in patients. While the majority of patients harbor two complete loss-of-function mutations in *NGLY1* (*He et al., 2015*), there are many symptoms such as seizures and scoliosis that are only reported in a subset of the patients (*Enns et al., 2014*). All patients experience developmental delay, but it ranges from slightly below average IQ to completely non-verbal (*Lam et al., 2017*). This variability based on background genetics was also observed in the lab when an NGLY1 deficiency mouse model was crossed onto an outbred mouse strain which partially rescued the lethality of the model (*Fujihira et al., 2017*). In order to identify components of the genetic background that affect the range of symptoms and severity of disease, we have utilized a collection of genetically diverse *Drosophila* strains known as the *Drosophila* Genetic Reference Panel (DGRP) (*Mackay et al., 2012*). By crossing a fly model of NGLY1 deficiency onto the panel, we recapitulated the variable phenotype seen in the human population. Here, we report the results of this cross and a list of candidate modifier genes derived from the genome-wide association (GWA) of the cross. To contextualize the candidate modifier list, we also performed an evolutionary rate covariation (ERC) analysis to identify genes that are co-evolving with *NGLY1*. Together these two genetic analyses have generated a list of genes that (1) may explain some of the variation seen between NGLY1 patients, (2) may encode proteins that physically interact with NGLY1 in ERAD or other cellular processes, and (3) may be direct deglycosylation targets of NGLY1. The top candidate modifier gene from the GWA is *NKCC1*, a conserved Na/K/Cl ion co-transporter. We found that *NKCC1* modifies multiple phenotypes in *Drosophila*, and in *NGLY1* -/- mammalian cells, NKCC1 displays abnormal average molecular weight and has reduced activity. The misregulation of NKCC1 likely explains several prominent secretory epithelium-related phenotypes observed in NGLY1 deficiency patients.

## Results

### Variation in lethality associated with NGLY1 deficiency

We crossed a fly model of NGLY1 deficiency (*Pngl* in flies, hereon referred to as *NGLY1*) onto 163 strains of the DGRP in order to assess the effect of natural variation on loss of *NGLY1* function. We have previously validated this NGLY1 deficiency model where an *NGLY1* RNAi reduces *NGLY1* transcript by >95% when driven by the ubiquitous *Tubulin*-GAL4 driver transgene (*Tubulin > NGLY1* RNAi) (*Owings et al., 2018*). In order to cross a ubiquitously expressed *NGLY1^RNAi* onto the DGRP strains in a single cross, we needed to overcome the lethality associated with loss of NGLY1 (*Owings et al., 2018*). To do this, a *Tubulin*-GAL80 transgene, which represses the effect of GAL4, was crossed onto the *Tubulin > NGLY1* RNAi background, such that RNAi is not expressed and flies from this parent strain are healthy and viable (*Figure 1A*). This donor strain was crossed to each DGRP strain to generate F1 flies that have both ubiquitous KD of *NGLY1* and 50% of their genome from each respective DGRP strain (*Figure 1B*). In this way, analyzing the F1 progeny was a direct measurement of the dominant effect of the DGRP genetic variants on the *NGLY1* KD phenotype.

The phenotypic outcome used for this screen was adult survival through eclosion. We simply scored all adults emerging from each cross in the four balancer categories: *CyO*, *Sb*, double balanced, or no balancers, with the no balancer flies being the *NGLY1* KD. If no lethality is present, Mendelian segregation should produce the expected 1:1:1:1 ratio of the genotypes. Given that there is a very low level of lethality associated with each balancer, the largest balancer class is the closest to the expected, and was used to calculate the ratio of lethality for *NGLY1* KD. Results of the screen reveal that survival to adulthood was strongly influenced by DGRP genetic background (*Figure 1B*; *Supplementary file 1*), with proportion of surviving flies ranging from 0.0 to 0.967. Survival to adulthood was not correlated with efficiency of RNAi, as there was no difference in KD efficiency in flies from either end of the phenotypic distribution (low surviving: 92.0% ± 3.7; high surviving: 94.8% ± 3.7; p=0.4). There was no correlation between proportion of surviving flies and the absolute number of flies in the balancer class ($R^2$ = 0.02; p=0.14), indicating that the ratio is not driven by the number of the balancer control flies.

### Genome-wide association

We hypothesized that the observed variable survival to adulthood in *NGLY1* KD flies was due to the underlying genetic variation in the DGRP. Therefore, genome-wide association (GWA) analysis of the fully sequenced DGRP was used to identify variants that associated with *NGLY1* KD survival. We used a linear mixed model to test 2,007,145 single-nucleotide polymorphisms (SNPs; *Supplementary file 2*). We recognize that our study suffers from a multiple

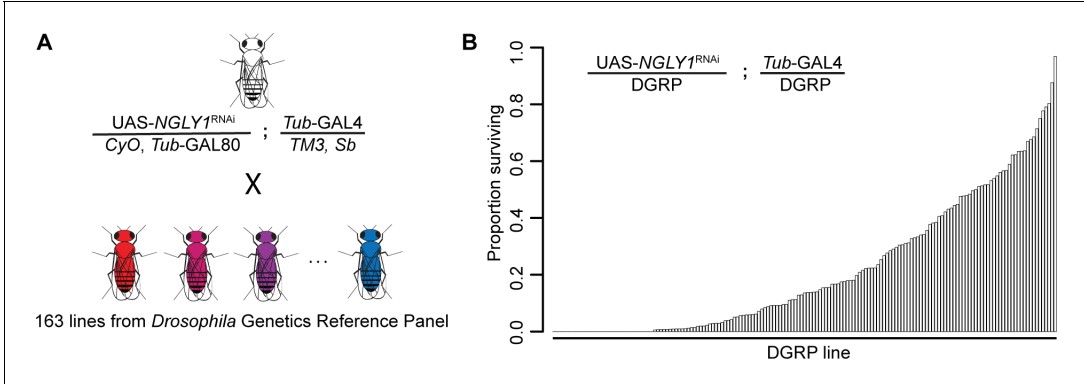

**Figure 1.** Lethality phenotype of *NGLY1* knockdown is highly modifiable by strain background. (**A**) *Drosophila* cross for *NGLY1* knockdown in each *Drosophila* genetic reference panel (DGRP) strain. (**B**) Proportion of *NGLY1* knockdown flies surviving for each cross was calculated based on the number eclosing compared to the expected number. Expected number was based on the largest control balancer class for each cross.
The online version of this article includes the following figure supplement(s) for figure 1:

**Figure supplement 1.** Ncc69 expression is not correlated with survival.

testing problem, making it difficult to interpret the role of any single SNP identified. Instead, the location of SNPs was used to identify candidate modifier genes. This type of approach has worked well in the past for other disease models (*Ahlers et al., 2019*; *Chow et al., 2013a*; *Chow et al., 2013b*; *Lavoy et al., 2018*; *Palu et al., 2019*) and provides an unbiased list of candidate genes that can be functionally tested for interactions with *NGLY1*.

At a nominal p-value of $p<10^{-5}$, 125 variants are associated with survival to adulthood. Of these 125 variants, 21 fall outside of a gene region (+/- 1 kb from the 5' or 3' UTRs) (*Supplementary file 3*). The remaining 104 variants map to 61 protein coding candidate genes (*Table 1*). Eighty-five of these 104 variants are in noncoding regions (UTRs, introns, or upstream or downstream) and 19 are in coding regions. Of these 19, 12 are synonymous changes and 7 are nonsynonymous (*exp*, *hiw*, *CG30048*, *SP2353*, *CG31690*, *Hrd3*, and *blue*). When multiple testing correction is applied to all the variants, the top 12 remain significant. Nine of these SNPs reside in an intron of the *Ncc69* gene. All nine SNPs are in strong linkage disequilibrium with each other, which is quite unusual for the DGRP. When we analyzed *Ncc69* expression levels using previously published RNAseq data from the DGRP (*Everett et al., 2020*), we found there was no correlation with survival (*Figure 1—figure supplement 1*).

Gene ontology (GO) enrichment analysis of the 61 candidate genes did not identify enrichment in any biological process or molecular function. However, GO enrichment was identified for the cellular component categories 'cell periphery' (GO:0071944; 19/61; q < 0.0016) and 'plasma membrane' (GO:0005886; 17/61; q < 0.004). At least 12/61 candidate genes are involved in protein homeostasis: three are involved in ERAD (*CG8405*, *CG42383*, and *Hrd3*), six are ER resident or membrane proteins (*CG33012*, *CG30043*, *CG31690*, *CG4341*, *Hrd3*, and *CG8405*), four are involved in ubiquitination or the proteasome (*hiw*, *blue*, *CG42383*, and *Hrd3*), one regulates heatshock responses (*Hsromega*), and one regulates N-linked glycosylation (*sff*).

Three of the identified ERAD genes already have known interactions with *NGLY1*. *CG8405* is the *Drosophila* ortholog of human *TMEM259*, which physically interacts with NGLY1 in co-immunoprecipitation experiments (*Zhu et al., 2017*). *CG42383* is the *Drosophila* ortholog of human *NSFL1C* (cofactor p47). NSFL1C and NGLY1 interact with the VCP/P97 AAA-ATPase complex involved in delivering misfolded proteins from the ERAD complex to the proteasome for degradation (*Kondo et al., 1997*; *McNeill et al., 2004*). *Hrd3* is the *Drosophila* ortholog of *SEL1L*. SEL1L is a component of the ERAD complex required for retrotranslocation of misfolded proteins from the ER to the cytoplasm for degradation. Recently, the *C. elegans* orthologs of *NGLY1* and *SEL1L* were both identified as modifiers of NRF1 function (*Lehrbach and Ruvkun, 2016*). These candidate genes are a proof-of-principle that this screen has identified functionally relevant modifiers.

The four candidate genes that encode ER resident proteins are particularly interesting. *CG31690* and *CG4341* are both *Drosophila* orthologs of human *TMTC2*, an ER transmembrane protein that regulates calcium homeostasis. *CG33012* and *CG30043* are both *Drosophila* orthologs of human *ERMP1*, an ER metalloprotease. It is striking that in both cases, both *Drosophila* orthologs of a single human gene were identified as candidate modifiers, suggesting that the function of *TMTC2* and *ERMP1* might be particularly important for *NGLY1* lethality. It is not clear how these genes might modify *NGLY1* lethality, but their physical localization to the ER makes sense and suggests a possible role in protein homeostasis as well.

## Gene set enrichment analysis

The rank-order candidate modifiers identified in our GWA ignores the majority of the association data by only considering one variant at a time, rather than all the variants associated with a particular gene. Therefore, we performed a gene set enrichment analysis (GSEA), which assigns each variant to the closest gene and generates a per gene metric for p-value enrichment (*Palu et al., 2019*; *Subramanian et al., 2005*). Given a defined set of genes annotated with a certain GO function, GSEA determines whether the members of that set are randomly distributed throughout the ranked list or if they are found primarily at the top or bottom of that list. We identified 21 gene sets positively associated with the ranked gene list (≥5 genes;>0.25 enrichment score; p<0.05) (*Figure 2*; *Supplementary file 4*). These data suggest that these GO categories are closely linked to NGLY1 activity and variation in individual genes in these categories contribute to the distribution of lethality observed in our screen.

**Table 1.** Candidate modifier genes identified from GWA.

Rank order of candidate genes was established based on the most significant associated SNP in the respective gene.

| Rank order | Gene | FBgn | Human ortholog | Periphery/membrane | Proteostasis |
|---|---|---|---|---|---|
| 1 | exp | FBgn0033668 | — | no | no |
| 2 | Ncc69 | FBgn0036279 | NKCC1/2 | yes | no |
| 3 | CG5888 | FBgn0028523 | — | yes | no |
| 4 | CG16898 | FBgn0034480 | — | no | no |
| 5 | bru3 | FBgn0264001 | CELF2/3/4/5/6 | no | no |
| 6 | CG31690 | FBgn0051690 | TMTC2 | no | yes |
| 7 | CG7227 | FBgn0031970 | SCARB1 | no | no |
| 8 | CR44997 | FBgn0266348 | — | no | no |
| 9 | rgn | FBgn0261258 | Many | no | no |
| 10 | M6 | FBgn0037092 | GPM6A | yes | no |
| 11 | Rab26 | FBgn0086913 | RAB26 | yes | yes |
| 12 | Obp56i | FBgn0043532 | — | no | no |
| 13 | 5-HT1A | FBgn0004168 | HTR1A | yes | no |
| 14 | CG33012 | FBgn0053012 | ERMP1 | no | yes |
| 15 | rst | FBgn0003285 | — | yes | no |
| 16 | CR43926 | FBgn0264547 | — | no | no |
| 17 | CG7337 | FBgn0031374 | WDR62 | no | no |
| 18 | hiw | FBgn0030600 | MYCBP2 | yes | yes |
| 19 | fid | FBgn0259146 | TRMT9B | no | no |
| 20 | nmo | FBgn0011817 | NLK | no | no |
| 21 | Sirup | FBgn0031971 | SDHAF4 | no | no |
| 22 | tst | FBgn0039117 | SKIV2L | no | no |
| 23 | Mdr50 | FBgn0010241 | many | yes | no |
| 24 | Cpr49Aa | FBgn0050045 | — | no | no |
| 25 | COX7C | FBgn0040773 | COX7C | no | no |
| 26 | Eip63E | FBgn0005640 | CDK14/15 | yes | no |
| 27 | CG30048 | FBgn0050048 | PKD1 | no | no |
| 28 | CG15040 | FBgn0030940 | — | no | no |
| 29 | SP2353 | FBgn0034070 | EGFLAM | no | no |
| 30 | Mf | FBgn0038294 | — | no | no |
| 31 | ome | FBgn0259175 | many | no | no |
| 32 | esn | FBgn0263934 | PRICKLE1-3 | no | no |
| 33 | haf | FBgn0261509 | many | no | no |
| 34 | dally | FBgn0263930 | GPC3/5 | yes | no |
| 35 | robo2 | FBgn0002543 | ROBO1/2/3/4 | no | no |
| 36 | Gyc32E | FBgn0010197 | NPR1/2 | yes | no |
| 37 | CG8170 | FBgn0033365 | many | no | no |
| 38 | CG8405 | FBgn0034071 | TMEM259 | no | yes |
| 39 | scaf | FBgn0033033 | — | yes | no |
| 40 | borr | FBgn0032105 | CDCA8 | yes | no |
| 41 | Syx7 | FBgn0267849 | STX7/12 | yes | no |
| 42 | DIP-delta | FBgn0085420 | many | yes | no |
| 43 | cv-c | FBgn0285955 | DLC1 | yes | no |

*Table 1 continued on next page*

Table 1 continued

| Rank order | Gene | FBgn | Human ortholog | Periphery/membrane | Proteostasis |
|---|---|---|---|---|---|
| 44 | Snmp2 | FBgn0035815 | CD36/SCARB1 | no | no |
| 45 | Mer | FBgn0086384 | NF2 | yes | no |
| 46 | sba | FBgn0016754 | — | no | no |
| 47 | Hsromega | FBgn0001234 | — | no | yes |
| 48 | CCAP-R | FBgn0039396 | NPSR1 | yes | no |
| 49 | Hrd3 | FBgn0028475 | SEL1L | no | yes |
| 50 | blue | FBgn0283709 | NEURL4 | no | yes |
| 51 | CG6262 | FBgn0034121 | TREH | no | no |
| 52 | CG45186 | FBgn0266696 | SVIL | no | no |
| 53 | Spn | FBgn0010905 | PPP1R9A | yes | no |
| 54 | dnc | FBgn0000479 | PDE4A/B/C/D | no | no |
| 55 | CG4374 | FBgn0039078 | many | no | no |
| 56 | sff | FBgn0036544 | many | no | yes |
| 57 | CG42383 | FBgn0259729 | NSFL1C | no | yes |
| 58 | Dyb | FBgn0033739 | DTNB | yes | no |
| 59 | CG34371 | FBgn0085400 | — | no | no |
| 60 | CG4341 | FBgn0028481 | TMTC2 | no | yes |
| 61 | CG30043 | FBgn0050043 | ERMP1 | no | yes |

Some of the most significantly enriched categories such as nuclear transport, rRNA processing and signal transduction are broad categories that could have wide reaching implications for NGLY1 function. These processes, however, are difficult to test and require long-term investigation, beyond the scope of this study. Circadian rhythm, on the other hand, is a specific and testable category. The enriched category for circadian rhythm function contains a number of genes that directly modulate circadian rhythm, including, *clock*, *period*, *timeless*, and *cycle*. We hypothesized that if variation in circadian rhythm function modifies lethality associated with loss of NGLY1 function then NGLY1 must affect the circadian rhythm. To test this, we knocked down *NGLY1* in the LNv pacemaker neurons in the central nervous system using the *Pdf*-GAL4 driver (*Renn et al., 1999*) and assayed rhythmicity of locomotor activity in constant darkness over 8 days in *Drosophila* Activity Monitors (DAM). Compared to *Pdf*-GAL4/+ and UAS-*NGLY1*^RNAi controls, flies with *NGLY1* KD exhibited a significantly longer period length (*Figure 2—figure supplement 1*), supporting the idea that NGLY1 function can affect sleep. Indeed it has been reported that patients with NGLY1 deficiency experience disturbed sleep patterns (*Enns et al., 2014*; *Lam et al., 2017*).

## Evolutionary rate covariation

Many of the GWA and GSEA results are intriguing, but appear far removed from the currently known functions of NGLY1. We hypothesized that we could contextualize some of the gene and network results by discovering which of them might be co-evolving with NGLY1. Therefore, we employed evolutionary rate covariation (ERC) analysis (*Wolfe and Clark, 2015*). Gene pairs with high ERC values have correlated rates of substitution and are thought to function together in protein complexes or related pathways. ERC analysis identified hundreds of protein-coding genes with integrated ERC scores exceeding two with *NGLY1* (column 'sumnlogpvbest' in *Supplementary file 5*). Of the 38 GWA candidates that have human orthologs, two were found in this group with elevated NGLY1 ERC values, *CG4374* (*many*) and *esn* (*PRICKLE1*). While this overlap is not enriched above background, co-evolution suggests that these two genes might have a particularly important interaction with *NGLY1*.

GO analysis was used to determine if there was enrichment in any biological pathways among *NGLY1* co-evolving genes. Among the top enriched pathways were 'rRNA/ncRNA/ribosome

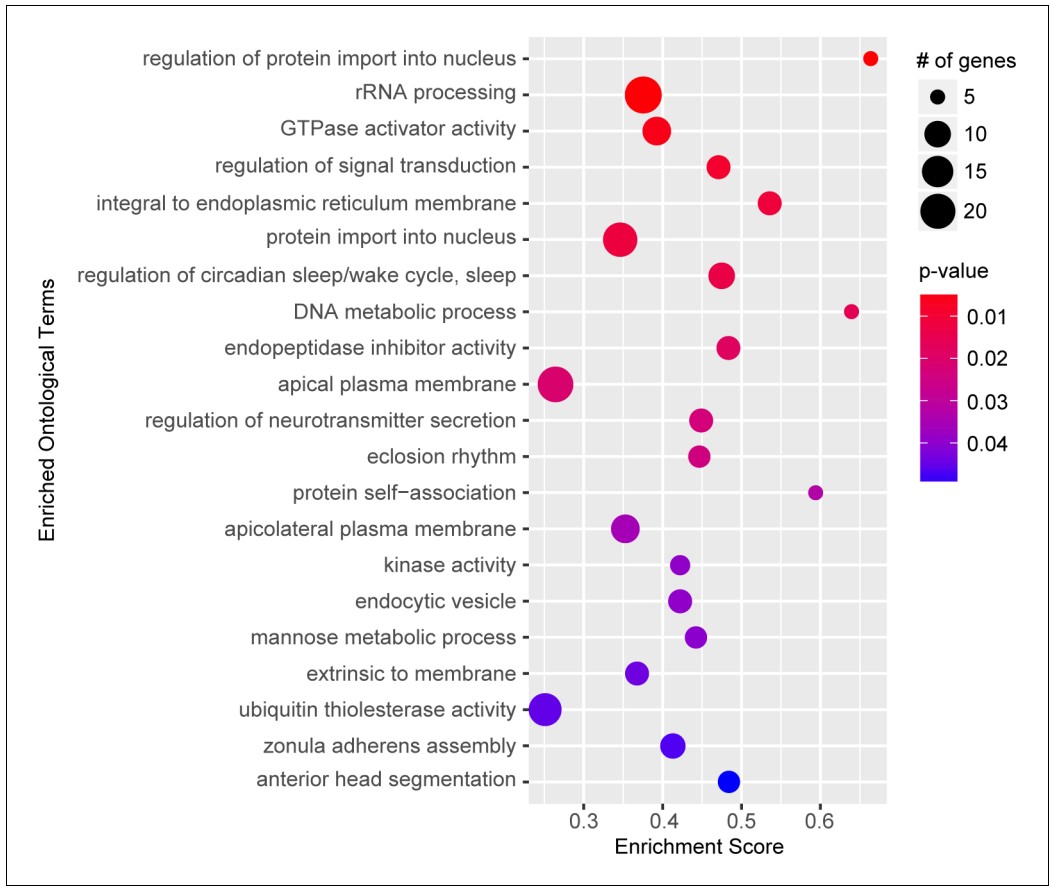

**Figure 2.** Gene set enrichment analysis. Top significant ontological categories identified by GSEA. p-values are indicated by red-to-blue gradient, with red the lowest p-values and blue the highest p-values. Gene number identified in each category is indicated by the size of the circle.

The online version of this article includes the following figure supplement(s) for figure 2:

**Figure supplement 1.** *NGLY1* knockdown causes circadian rhythm defect.

biogenesis/metabolism-related functions' and 'functions related to nuclear pore complex'. This is particularly exciting as both processes overlap with the top GO enrichment categories observed in the GSEA analysis, suggesting that the same functional categories that contribute to variation in *NGLY1*-related lethality also appear to contain genes that co-evolve with *NGLY1*. The rRNA processing category (GO:0006364) contained six genes overlapping between the two analyses. This overlap is higher than expected, given two equally sized random groups of genes (GSEA: 23 genes; ERC: 37 genes; $p<2.6\times10^{-12}$). Among other ncRNA-related enriched GO categories from the ERC analysis are ncRNA metabolic process (GO:0034660), ncRNA processing (GO:0034470), tRNA metabolic process (GO:0006399), ribosome biogenesis (GO:0042254), and tRNA modification (GO:0006400). The functions related to the nuclear pore included nuclear export (GO:0051168), nuclear pore organization (GO:0006999), nuclear transport (GO:0051169), and nuclear pore complex assembly (GO:0051292). While there was no overlap between ERC and GSEA for exact nuclear pore function categories, GSEA results were enriched for functions related to nuclear import (GO:0042306 and GO:0006606). Together, these observations suggest previously unknown roles for NGLY1 in ncRNA and nuclear pore functions.

NGLY1 deficiency is part of a larger category of disorders known as Congenital Disorders of Glycosylation (CDG), with NGLY1 being the only protein that actually *de*glycosylates substrates. There are 151 known CDG genes. GO analysis of the ERC results identified enrichment of the GPI anchor biosynthetic process, which contains several of these CDG genes, leading us to believe that other CDG genes may have been ERC hits. However, the 151 CDG genes do not fall into one functional

GO category, therefore, we manually curated the ERC list and identified 26 CDG genes that co-evolve with NGLY1 (see color coding in *Supplementary file 5*). This represents a significant overlap above what is expected by chance ($p<7.6\times10^{-10}$). In particular, 5 of the 21 genes involved in N-linked glycosylation and 9 of the 29 genes involved in GPI-anchor biogenesis are co-evolving with *NGLY1*. The remaining 10 genes are spread across the CDG functional spectrum. The identification of a number of CDG genes that co-evolve with *NGLY1*, suggests that NGLY1 function might be important to the broader glycosylation pathways.

## Genetic interaction between *NGLY1* and *Ncc69* in *Drosophila*

While these genetic analyses revealed many promising modifying and co-evolving genes which should be investigated, we began by investigating *Ncc69* because it was the top hit with a human ortholog in our GWA analysis. Further, Ncc69 is a glycoprotein, making it a potential target of NGLY1 deglycosylation. *Ncc69* has two mammalian orthologs, *NKCC1* and *NKCC2*. While *Ncc69* is ubiquitously expressed in *Drosophila*, NKCC1 (gene: *SLC12A2*) is highly expressed in secretory epithelia and NKCC2 (gene: *SLC12A1*) is primarily expressed in the kidney (*Delpire and Gagnon, 2018*). In all cases, the protein is a 12-pass transmembrane cation-chloride co-transporter that brings Na$^+$, K$^+$, and Cl$^-$ into the cell (*Delpire and Gagnon, 2018*). Mutations in *NKCC2* are known to cause type I Bartter syndrome (*Simon and Lifton, 1996*) and a recent clinical report shows homozygous loss-of-function mutations in *NKCC1* cause the novel disease Kilquist syndrome (*Macnamara et al., 2019*).

To validate the genetic interaction observed between *NGLY1* and *Ncc69* in the GWA, we generated ubiquitous double knockdown (DKD) *Drosophila* and scored offspring that survived to eclosion (*Figure 3A*). The fraction of KD flies was calculated from observed offspring of the balancer phenotype. *NGLY1* KD caused a decrease in survival to ~25%, in accordance with our previous report of this RNAi line (*Owings et al., 2018*). *Ncc69* KD did not cause any significant decrease in survival ($\chi^2=1.002$, p=0.3168), as previously reported (*Leiserson et al., 2011*). The DKD, however, caused complete lethality. This synthetic lethality confirms *Ncc69* as a hit from the *NGLY1* modifier screen.

KD of cation-chloride cotransporters in glia has been shown previously to cause seizures in *Drosophila* (*Rusan et al., 2014*), and we wanted to test whether this phenotype could be modified by *NGLY1* KD. We performed single and double knockdowns of *NGLY1* and *Ncc69* in glial cells using the *repo*-GAL4 driver. This *Ncc69* RNAi line is the same as the one described above. *Drosophila*

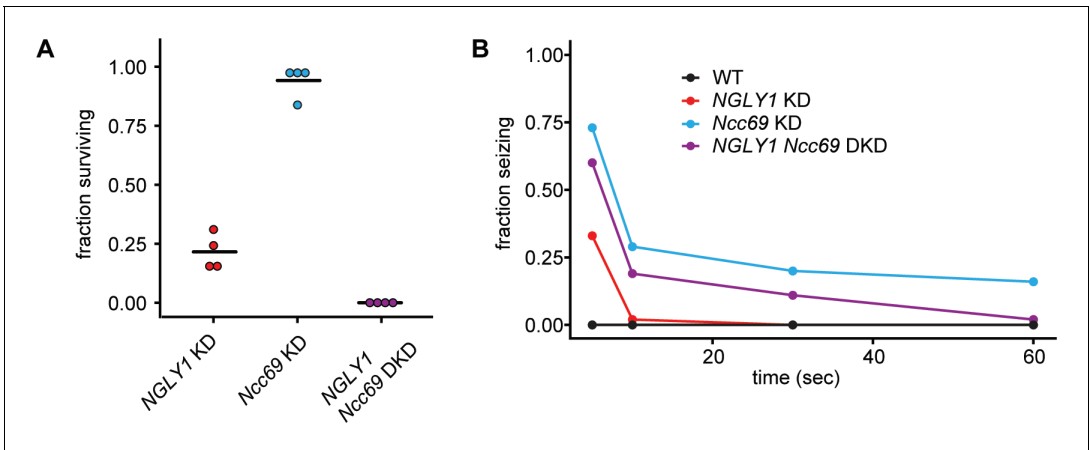

**Figure 3.** *NGLY1* and *Ncc69* interact genetically in *Drosophila*. (**A**) Proportion of flies surviving to eclosion in ubiquitous knockdowns. *NGLY1* knockdown (KD) are UAS-*Pngl*^RNAi/+; *Tubulin*-GAL4/+. *Ncc69* KD are UAS-*Ncc69*^RNAi/+; *Tubulin*-GAL4/+. *NGLY1 Ncc69* double knockdown (DKD) are UAS-*Pngl*^RNAi/+ UAS-*Ncc69*^RNAi/*Tubulin*-GAL4/+. Four separate matings were performed for each cross with at least 40 offspring generated for the balancer control for each. Fraction surviving is calculated compared to balancer offspring. Chi-square analysis was performed for the total number of flies compared to expected Mendelian numbers. NGLY1 KD $\chi^2$ = 109.7, p<0.0001; Ncc69 KD $\chi^2$ = 1.002, p=0.3168, and NGLY1 Ncc69 DKD $\chi^2$ = 186, p<0.0001. (**B**) Bang sensitivity assay to assess seizures in glial knockdown flies. WT flies are attP2 and attP40. *NGLY1* KD are UAS-*Pngl*^RNAi/+; *repo*-GAL4/+. *Ncc69* KD are UAS-*Ncc69*^RNAi/+; *repo*-GAL4/+. *NGLY1 Ncc69* DKD are UAS-*Pngl*^RNAi/+; UAS-*Ncc69*^RNAi/ *repo*-GAL4. For each genotype, at least 45 4- to 7-day-old females were used to calculate the percent seizing at a given time after vortexing. Repeated measures ANOVA p-value=0.000176.

were assessed for seizure phenotype using the bang sensitivity assay (*Figure 3B*). Control, wild-type flies show immediate recovery, as expected. In *NGLY1* KD flies, ~30% showed severe seizures in the form of complete immobility 5 s following vortex. However, by 10 s following vortex, *NGLY1* KD flies were completely recovered. This is the first report of seizure phenotype in any NGLY1 deficiency model, mimicking what is observed in patients. *Ncc69* KD flies showed severe seizures with 75% seizing at 5 s following vortex, in line with previous reports (*Rusan et al., 2014*). In the DKD, there was a partial rescue of the severe *Ncc69* phenotype. At all time-points between 5 and 60 s, the DKD flies showed an intermediate phenotype relative to *NGLY1* and *Ncc69* single KDs, confirming a genetic interaction between *NGLY1* and *Ncc69*.

### Functional analysis of NKCC1 in *NGLY1* null MEFs

To understand the cell biology behind the genetic interaction that was observed in *Drosophila*, we utilized *NGLY1* knockout (-/-) mouse embryonic fibroblasts (MEFs) (jax.org/strain/027060). Fibroblasts should only express the ubiquitous ortholog, NKCC1 (*Haas and Forbush, 1998*). When the membrane fraction of *NGLY1* -/- MEFs was analyzed by immunoblot for NKCC1 there was a noticeable shift in the average molecular weight of the band compared to wildtype, control cells (+/+) (*Figure 4A*). Using the molecular weight marker to calculate the size of the proteins (un-cropped blot in *Figure 4—figure supplement 1*), the upper limits of the bands were ~170 kDa for both +/+ and -/- cells (*Figure 4B*). The lower limit of the bands, however, were ~140 kDa for +/+ cells and ~150 kDa for the -/- cells (*Figure 4C*).

To determine if a glycosylation event might be responsible for this size difference, cell lysates were treated with deglycosylating enzymes (*Figure 4D*). PNGase F removes all N-linked glycans and this treatment caused a large decrease in the molecular weight, to ~125 kDa in NKCC1 proteins from both *NGLY1* +/+ and -/- cells. The expected weight of mouse NKCC1 without any post-translational modifications is 130 kDa indicating that all post-translational modifications are likely N-linked

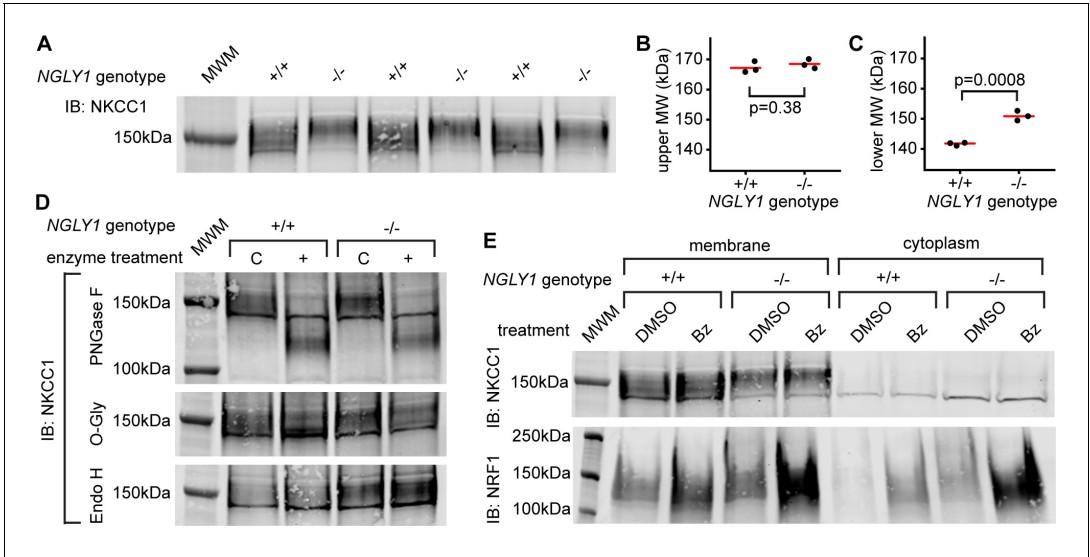

**Figure 4.** Endogenous NKCC1 is altered in NGLY1-deficient MEFs. (**A**) Control (+/+) and *NGLY1* null (-/-) MEFs were grown to confluency and then lysed to isolate the membrane and cytoplasmic fractions. Three separate membrane lysates for both genotypes were analyzed by immunoblotting for NKCC1 compared to a molecular weight marker (MWM). Blot was used for molecular weight calculations of the upper-most limit (see un-cropped blot in *Figure 4—figure supplement 1*) (**B**) And the lower-most limit (**C**) of the protein band. Red bar represents the mean. Two-tailed t-test was used to calculate p-values. (**D**) Membrane lysates from MEFs were treated with N-Glycosidase F (PNGase F), O-Glycosidase (O-Gly), or Endoglycosidase H (Endo H) for 1 hr then analyzed by immunoblot. Control (C)Samples were treated in all the same conditions but without the added enzyme. (**E**) MEFs were treated with 500 nM bortezomib (Bz) or equal volume of vehicle control (DMSO) for 4 hr then lysed to collect membrane and cytoplasmic fractions. Lysates were analyzed by immunoblotting for NKCC1. NRF1 was analyzed as a positive control of proteasome inhibition. Dark band at about 140 kDa in both (**A**), (**D**), and (**E**) is believed to be non-specific.

The online version of this article includes the following figure supplement(s) for figure 4:

**Figure supplement 1.** Full western blot of NKCC1 in NGLY1 +/+ and -/- MEFs.

glycans. This is in accordance with the prediction of two canonical N-linked glycosylation sites (*Payne et al., 1995*). The fact that there is no difference in molecular weight between the +/+ and -/- after treatment indicates the difference observed in the untreated state was eliminated by the PNGase enzyme. Treatment with O-Glycosidase had no effect on the molecular weight of the band in either the +/+ or -/- lysates. Although O-Glycosidase does not cleave every type of O-linked glycan, these results coupled with the PNGase results indicate there are likely no O-linked glycans on NKCC1. Finally, to determine the maturation state of the N-linked glycans Endoglycosidase H was used. No change in molecular weight was seen, indicating the N-linked glycans are no longer in the high-mannose state in both the +/+ and -/- cells.

The most well-studied substrate of NGLY1, NRF1, is degraded by the proteasome under normal conditions, and only when the proteasome is stressed or inhibited does NRF1 become active. Indeed, NGLY1 is thought to act in the ERAD pathway and therefore all its substrates may be regulated in some way by proteasomal degradation. To test if NKCC1 abundance is affected by the proteasome, MEFs were treated with the proteasome inhibitor bortezomib (Bz). As expected, there was an increase in NRF1 abundance during proteasome inhibition (*Figure 4E*). However, there was no noticeable increase in NKCC1 protein for either the +/+ or -/- MEFs, indicating proteasomal degradation is not a major regulator of NKCC1.

Given the altered glycosylation state of NKCC1 observed in *NGLY1* -/- MEFs, we wanted to determine the functionality of NKCC1 in these cells. Previous reports have shown that inhibiting N-linked glycosylation can decrease functionality for both NKCC1 (*Singh et al., 2015*) and for NKCC2 (*Paredes et al., 2006*), and therefore, we hypothesized that the misglycosylation might also decrease function. The NKCC proteins and the Na$^+$/K$^+$-ATPase can both transport Rb$^+$ in place of K$^+$, so we incubated cells with radioactive $^{86}$Rb and measured cellular uptake. Cells were assayed in the presence or absence of 10 µM bumetanide, an NKCC inhibitor, or 100 µM ouabain, a Na$^+$/K$^+$-ATPase inhibitor, as compared to vehicle control. When the ouabain-sensitive and bumetanide-sensitive activities were summed, they accounted for all of the $^{86}$Rb flux observed (*Figure 5—figure supplement 1*). We found that bumetanide-sensitive $^{86}$Rb flux, but not ouabain-sensitive flux, was impaired by ~50% in the -/- MEFs, indicating a specific defect in NKCC1 activity without impairment in the Na$^+$/K$^+$-ATPase (*Figure 5*). We assayed ion transport activity under three conditions of isotonic, hypertonic, or hypotonic baths. Although both hypertonic and hypotonic low chloride baths can stimulate NKCC1 activity in other cell types (*Darman and Forbush, 2002*), we did not see any

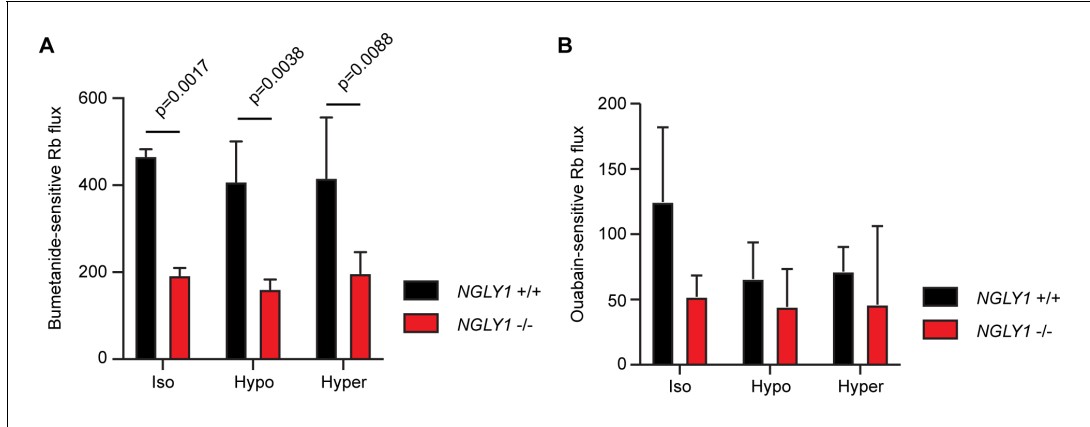

**Figure 5.** *NGLY1* -/- MEFs show decreased NKCC1-specific ion flux. (**A**) Bumetanide-sensitive $^{86}$Rb flux was measured in *NGLY1* +/+ and *NGLY1* -/- MEFs to measure NKCC1 activity. Flux was examined in three bath conditions, isotonic (iso), hypotonic (hypo), and hypertonic (hyper). There was a significant effect of genotype (p<0.0001) in two-way ANOVA, with no significant effect of condition (p=0.5756) or interaction (p=0.8075). Adjusted p-values for Sidak's multiple comparisons test between *NGLY1* +/+ and *NGLY1* -/- are shown in the figure. (**B**) Ouabain-sensitive $^{86}$Rb flux was measured in *NGLY1* +/+ and *NGLY1* -/- MEFs to measure Na$^+$/K$^+$-ATPase activity in the same three conditions as in A. There were no significant effects of genotype (p=0.0516), condition (p=0.3047) or interaction (p=0.4711) by two-way ANOVA, indicating the *NGLY1* knockout has a specific effect on NKCC1 activity without affecting Na$^+$/K$^+$-ATPase activity.

The online version of this article includes the following figure supplement(s) for figure 5:

**Figure supplement 1.** $^{86}$Rb uptake in MEFs occurs through bumetanide-sensitive and ouabain-sensitive pathways.

effect of bathing medium in the MEFs, consistent with a recent report of lack of hypertonic stimulation of NKCC1 in human fibroblasts (*Delpire et al., 2016*). Together, these data demonstrate that loss of NGLY1 results in a change in the glycosylation state of NKCC1 and a significant reduction in NKCC1 function.

## Discussion

Like many rare diseases, research into the pathogenesis of NGLY1 deficiency has been narrowly focused, based on early hypotheses. This often limits how we understand the connection between a particular disease and other pathways. For NGLY1 deficiency specifically, basic research and potential therapies have focused intensely on NRF1, the first well-established substrate of NGLY1 deglycosylation. Motivated by the extensive phenotypic variation among NGLY1 deficiency patients, we took advantage of natural genetic variation in *Drosophila* to identify modifiers of NGLY1 deficiency. This unique screen demonstrated that (1) we can model the extensive phenotypic variation observed and (2) that genetic variation can cause this phenotypic variability. Association analysis then identified a number of exciting candidate modifier genes. Here, we have validated the novel and conserved modifier *NKCC1* (*Drosophila Ncc69*), a new potential therapeutic target for NGLY1 deficiency.

A major advantage of screens is the identification of previously unanticipated biological connections. First, our association analysis of the lethality screen has generated a list of 61 genes that we hope the scientific and patient communities will be able to use. Second, GSEA identified several pathways, including rRNA metabolism and nuclear transport that are surprising based on known NGLY1 functions. Third, ERC analysis identified genes that are coevolving with NGLY1 across the animal kingdom, including both rRNA/ncRNA pathways and nuclear transport. It appears that rRNA metabolism and nuclear transport are likely important to NGLY1 function, yet it remains unclear how NGLY1 is connected to these pathways. Components of the ribosome and the nuclear pore are often O-glycosylated. While there is no direct connection between NGLY1 and O-GlcNAcylation, we have previously demonstrated that loss of NGLY1 impacts UDP-GlcNAc levels (*Owings et al., 2018*). It is highly plausible that a misregulation of UDP-GlcNAc levels could affect O-GlcNAcylated proteins. More work is needed to determine exactly how NGLY1 is connected to these unexpected pathways. Finally, the ERC analysis also identified 26/151 known Congenital Disorders of Glycosylation (CDG) genes. While NGLY1 is also classified as a CDG, it is unclear why there might be co-evolution with other CDG genes. Perhaps, there is a feedback mechanism, again, related to UDP-GlcNAc biosynthesis that connects these genes. These results suggest that there is a previously unknown connection between these loosely connected CDG genes.

When analyzing the list of modifier genes, it is apparent that many of the candidates are involved in ERAD. This offers a proof-of-principle that this screen is well suited for identifying bona fide biologically relevant modifiers. Several previous studies linked NGLY1 with the ERAD process (*Bebök et al., 1998*; *Katiyar et al., 2005*; *Park et al., 2001*). Yet, NGLY1 does not appear to be required for proper ERAD function (*Hirsch et al., 2003*; *Misaghi et al., 2004*). While perturbations to ERAD often result in ER stress, we have previously reported that there was no functional or transcriptome evidence for ER stress in a *Drosophila* model of NGLY1 deficiency (*Owings et al., 2018*). Others have reported no ER stress in NGLY1 -/- human cells, mice, and rats (*Asahina et al., 2020*; *Mueller et al., 2020*; *Tambe et al., 2019*). However, there is conflicting evidence for ER stress as it was recently reported that ER stress markers were upregulated in NGLY -/- MEFs (*Galeone et al., 2020*). Nevertheless, in our current screen, we did not identify any genes involved in canonical ER stress responses, suggesting that ER stress might not play a large role in the pathogenesis of the disease. Here, we have reported that NKCC1 is altered in NGLY1 -/- cells, however we found that inhibiting the proteasome had no effect on the protein abundance. Thus, it may be that NGLY1 functions to regulate proteins in various ways that are closely related to ERAD, but that do not result in direct proteasomal degradation, and thereby, do not cause an accumulation of misfolded protein and ER stress.

As with most hypothesis-free approaches, our analyses produced many exciting new avenues for exploration. In order to keep our work relevant and translatable to the clinic, we have focused our follow up experiments on genes with human orthologs. In this report we began with our top hit, *Ncc69* (human *NKCC1/2*), which encodes an SLC12 $Na^+/K^+/2Cl^-$ transporter. In cells derived from an

NGLY1 -/- mouse model, we found that NKCC1 protein migrated at a higher average molecular weight relative to +/+ cells. We found this altered glycosylation was accompanied by a ~ 50% reduction in NKCC1 activity.

NKCC1 contains two canonical N-linked glycosylation sites (*Payne et al., 1995*) similar to those validated in NKCC2 (*Paredes et al., 2006*). As expected, both sites are in an extracellular loop, and therefore, these sites face the lumen of the ER during protein translation and maturation. Because of this arrangement, these two sites are not predicted to be accessible by NGLY1, which is localized to the cytoplasm. However, recent work indicates that NGLY1 can deglycosylate proteins prior to their complete retrotranslocation out of the ER (*Galeone et al., 2020*). NGLY1 is recruited to the cytosolic surface of the ER where it deglycosylates BMP4, which is in fact necessary for its retrotranslocation to the cytoplasm. Perhaps NKCC1 is being deglycosylated on one of its two canonical N-linked glycosylation sites through a similar mechanism. Alternatively, it may be that NGLY1 is acting on a non-canonical, cytoplasmic N-linked glycosylation site. Sequence analysis reveals three other asparagine residue within the necessary N-X-S/T sequence for N-linked glycosylation, however, two are predicted to be in transmembrane domains; the third (human NKCC1 residue N168) is located in the amino-terminal cytoplasmic tail of the protein. The recent Cryo-EM structure determined that the amino-terminal tail is disordered and that the carboxy-terminal tail acts as a regulatory domain (*Chew et al., 2019*). Although rare, there have been reports of cytosolic N-linked glycosylation, including on the dog kidney Na+, K+-ATPase pump (reviewed in *Hart et al., 2017*). In all these cases, and in our case here, the mechanism for cytoplasmic glycosylation remains unknown. Determining if one of the two canonical sites is altered or if a non-canonical, cytoplasmic site is altered is a top priority for future work.

While we observe a glycosylation difference on NKCC1, and NGLY1 is a deglycosylating enzyme, we cannot eliminate the possibility that this NKCC1 regulation is a secondary effect. NGLY1 may be regulating an intermediary protein that in turn, regulates NKCC1. This was recently found to be the case for aquaporins in NGLY1-deficient cells (*Tambe et al., 2019*). NGLY1 was found to regulate the abundance of transcription factors Atf1/Creb1 independent of its enzymatic activity. Atf1/Creb1 then, in turn, regulates the transcription of several aquaporin subunits. Given that we show the abundance of NKCC1 does not change, but rather NKCC1 has a molecular weight shift, it is likely that the altered state is due to some difference in a post-translational modification. If this effect is secondary, it may be that NGLY1 is directly affecting Golgi-localized glycosyltransferases that in turn modify the already present glycans. Or that NGLY1 is altering the function of a sialyltransferase, thereby altering sialic acid residues on NKCC1 which are known to cause significant changes in migration on SDS-PAGE. This hypothesis is supported by the fact that Endo H treatment did not affect NKCC1 from either +/+ or -/- cells, indicating that NKCC1 has been fully processed through the Golgi. Future work to test this hypothesis could entail mass spectrometry analysis to determine the specific glycan structures on NKCC1.

Identification of targets and modifier genes should provide insight into the pathogenesis of a disease and help explain some of the patient phenotypes. When NRF1 was identified as the first target of NGLY1, it provided insight into some of the molecular defects observed in NGLY1-deficient cells, including deficits in proteasomal function and expression. These cellular phenotypes, however, did not translate well into insight into the complex patient symptoms. In contrast, decreased NKCC1 activity may explain some of the prominent features of NGLY1 deficiency. NKCC1 functions in many secretory epithelia, such as the salivary, sweat, and lacrimal glands, to allow basolateral ion uptake and subsequent secretion (*Delpire and Gagnon, 2018*). Therefore, a decrease in NKCC1 activity could well explain the alacrima and reduced saliva and sweat production seen in NGLY1 deficiency. Strikingly, a recent clinical report describes a patient with a homozygous deletion in *NKCC1* (null) who has many overlapping features with NGLY1 deficiency patients, including absence of saliva, tears, and sweat (Kilquist syndrome) (*Macnamara et al., 2019*). The *NKCC1* null mouse also displays defects in salivation (*Evans et al., 2000*). Other notable, but perhaps less specific, features in the NKCC1-deficient child, including developmental delay and gastrointestinal problems, also overlap with those observed in NGLY1 deficiency. The NKCC1-deficient patient also had severe hearing loss, cochlear defects, and abnormal auditory brainstem responses (ABRs). Several patients with missense mutations in NKCC1, rather than complete loss-of-function mutations, display bilateral sensorineural hearing loss (*McNeill et al., 2020*). NGLY1 deficiency patients do not have severe hearing loss, but do report abnormal ABRs. This difference may be explained by a 50% reduction in NKCC1 activity,

rather than complete loss of activity. The overlap in a majority of the symptoms between this new syndrome and NGLY1 deficiency strengthens the case for both a genetic and functional connection between NGLY1 and NKCC1.

NKCC1 may be a promising target for the development of NGLY1 deficiency therapies. Given that it is a transporter and partially exposed to the extracellular space, NKCC1 could be particularly amenable to modulation by small molecules. Based on our work reported here, we predict that increasing function of NKCC1 may ameliorate some symptoms. Quercetin, a flavonoid, is a readily available molecule that has been shown to enhance NKCC1 activity (*Asano et al., 2009*; *Nakajima et al., 2011*). These studies demonstrated that quercetin significantly increased $^{86}$Rb uptake in cell culture and that this increase was bumetanide-sensitive, indicating specificity to NKCC1. This is similar to other studies showing flavonoids increasing the activity of channels, such as the flavonoid genistein increasing activity of the cystic fibrosis transmembrane conductance regulator (CFTR) (*Sugawara and Nikaido, 2014*), and several flavonoids targeting cardiovascular channels (*Scholz et al., 2010*). These quercetin studies, however, are in the context of normal functioning NKCC1 protein. In our study, NKCC1 is altered from the WT state and it is unknown whether quercetin would be able to modulate the activity of an altered NKCC1. Strikingly, however, quercetin was recently discovered in a drug screen to provide benefit to NGLY1-deficient *C. elegans* (*Iyer et al., 2019*). These results coupled with our discovery of NKCC1 as a NGLY1 substrate offer an exciting new avenue of treatment for NGLY1 deficiency patients. Targeted studies are needed to determine if quercetin or other molecules could specifically enhance NKCC1 function in the context of NGLY1 deficiency.

In this study, we took a series of unbiased approaches in *Drosophila* to identify modifiers of NGLY1 deficiency. This resulted in a number of new insights into the potential pathogenesis of NGLY1 that we hope others will also investigate. With rare diseases like NGLY1 deficiency, unbiased and forward genetic approaches are an efficient method for expanding possible avenues of investigation and therapeutic development. This study also highlights the power of using model organisms like *Drosophila* to uncover pathways and genes that can be validated in mammalian systems and targeted for therapeutic development.

## Materials and methods

**Key resources table**

| Reagent type (species) or resource | Designation | Source or reference | Identifiers | Additional information |
|---|---|---|---|---|
| Gene (*Drosophila melanogaster*) | *Pngl* | GenBank | ID:35527 Dmel_CG7865 | |
| Gene (*Drosophila melanogaster*) | *Ncc69* | GenBank | ID: 39410 Dmel_CG4357 | |
| Gene (*Mus musculus*) | *NGLY1* | GenBank | ID: 59007 | |
| Gene (*Mus musculus*) | *NKCC1* | GenBank | ID: 20496 | *Slc12a2* |
| Genetic reagent (*Drosophila melanogaster*) | *Pngl*-RNAi | Bloomington *Drosophila* Stock Center | RRID:BDSC_54853 | y1 v1; P{y+t7.7 v+t1.8=TRiP. HMJ21590}attP40 |
| Genetic reagent (*Drosophila melanogaster*) | Tubulin-GAL4 | Bloomington *Drosophila* Stock Center | RRID:BDSC_5138 | y1 w*; P{w+mC = tubP-GAL4}LL7/TM3, Sb1 Ser1 |
| Genetic reagent (*Drosophila melanogaster*) | Tubulin-GAL80 | Bloomington *Drosophila* Stock Center | RRID:BDSC_5190 | y1 w[*]; P{w[+mC]=tubP-GAL80}LL9 P{w[+mW.hs]=FRT(w[hs])}2A/TM3, Sb1 |

*Continued on next page*

*Continued*

| Reagent type (species) or resource | Designation | Source or reference | Identifiers | Additional information |
|---|---|---|---|---|
| Genetic reagent (*Drosophila melanogaster*) | *Drosophila* Genetics Reference Panel | Bloomington *Drosophila* Stock Center | Set of 194 strains, example strain: BDSC:55014, RRID:BDSC_55014 | |
| Genetic reagent (*Drosophila melanogaster*) | Pdf-GAL4 | Bloomington *Drosophila* Stock Center | RRID:BDSC_6899 | P{w[+mC]=Pdf-GAL4.P2.4}X, y1 w[*] |
| Genetic reagent (*Drosophila melanogaster*) | UAS-Pngl-RNAi | Bloomington *Drosophila* Stock Center | RRID:BDSC_42592 | y1 sc* v1 sev21; P{y+t7.7 v+t1.8=TRiP. HMS02424}attP40 |
| Genetic reagent (*Drosophila melanogaster*) | UAS-Ncc69-RNAi | Bloomington *Drosophila* Stock Center | RRID:BDSC_28682 | y1 v1; P{y+t7.7 v+t1.8=TRiP. JF03097}attP2 |
| Cell line (*Mus musculus*) | Mouse embryonic fibroblasts (MEF) | Jackson Labs | Primary line from mouse #027060, https://www.jax.org/strain/027060 | |
| Antibody | Anti-NKCC1 (Rabbit polyclonal) | Cell Signaling | Cat#14581, RRID:AB_2798524 | IB: 1:1000 |
| Antibody | Anti-TCF11/NRF1 (Rabbit monoclonal) | Cell Signaling | Cat#8052, RRID:AB_11178947 | IB: 1:1000 |
| Antibody | IRDye 800CW Goat-anti-rabbit | Abcam | Cat#216773 | IB: 1:10,000 |
| Commercial assay or kit | Cell Fractionation Kit | Cell Signaling | Cat#9038 | |
| Chemical compound, drug | Bortezomib | EMD Millipore | Cat# 179324-69-7 | |
| Software, algorithm | Genome Wide Association | *Chow et al., 2016* | | |
| Software, algorithm | Gene Set Enrichment Analysis | *Subramanian et al., 2005* | | |
| Software, algorithm | Evolution Rate Covariation | *Clark et al., 2012* | | |
| Software, algorithm | R | https://www.r-project.org/ | | |

## *Drosophila* lines

Flies were maintained at 25°C on a 12 hr light/dark cycle and raised on a standard diet based on the Bloomington Stock Center standard medium with malt. All flies were aged 3–5 days old for experiments. For the DGRP screen, the following *D. melanogaster* stocks were used: *Pngl*[RNAi] (Bloomington *Drosophila* Stock Center: 54853) and *Tubulin*-GAL4 driver (5138). The *Tubulin*-GAL80 strain was provided by Dr. Carl Thummel (University of Utah). The DGRP strains are available at the Bloomington *Drosophila* Stock Center. To measure circadian rhythm, the following stocks were used: a *w*- Berlin control strain, a *w; Pdf*-GAL4 strain (outcrossed to w- Berlin), and a *yv*; UAS-*Pngl*[RNAi] strain (Bloomington stock center #42592). For bang sensitivity assays, the following stocks were used: UAS-*Pngl*[RNAi] (BL #54853), UAS-*Ncc69*[RNAi] (BL #28682), and *repo*-GAL4. These stocks were obtained from the Bloomington Stock Center and Adrian Rothenfluh (University of Utah) respectively.

## DGRP screen

Virgin females from the DGRP strains were fed yeast overnight and then crossed with males from the donor strain UAS-*Pngl*[RNAi]/*Cyo*,*Tubulin*-GAL80; *Tubulin*-GAL4/TM3,*Sb* in two replicate bottles.

Progeny were collected and scored for the four balancer classes: *CyO*, *Sb*, double balanced, or no balancers, with the no balancer flies being the *NGLY1* KD. This cross should produce the expected 1:1:1:1 ratio of the four genotypes. Given that there is always a very low level of lethality associated with each balancer, the largest balancer class was considered the closest to the expected number. We scored at least 200 flies per DGRP cross. Males and females were combined for a single count. To calculate the proportion of *NGLY1* KD flies by generating a ratio of *NGLY1* knockdown/largest balancer class. This metric was used for the GWA.

## Genome wide association

GWA was performed as previously described (*Chow et al., 2016*). DGRP genotypes were downloaded from the website, http://dgrp.gnets.ncsu.edu/. Variants were filtered for minor allele frequency ($\geq$0.05), and non-biallelic sites were removed. A total of 2,007,145 variants were included in the analysis. The proportion of *NGLY1* KD flies surviving was regressed on each SNP. To account for cryptic relatedness (*He et al., 2014*; *Huang et al., 2014*), GEMMA (v. 0.94) (*Zhou and Stephens, 2012*) was used to both estimate a centered genetic relatedness matrix and perform association tests using the following linear mixed model (LMM):

$$y = \alpha + x\beta + u + \epsilon$$
$$u \sim \text{MVN\_n}(0, \lambda\, \tau^{\wedge}(-1)\text{K})$$
$$\varepsilon \sim \text{MVN\_n}(0, \tau^{\wedge}(-1)\text{I\_n})$$

where, as described and adapted from *Zhou and Stephens, 2012*, y is the n-vector of proportion lethality for the n lines, $\alpha$ is the intercept, x is the n-vector of marker genotypes, $\beta$ is the effect size of the marker. u is a n x n matrix of random effects with a multivariate normal distribution (MVN_n) that depends on $\lambda$, the ratio between the two variance components, $\tau(-1)$, the variance of residuals errors, and where the covariance matrix is informed by K, the calculated n x n marker-based relatedness matrix. K accounts for all pairwise non-random sharing of genetic material among lines. $\epsilon$, is a n-vector of residual errors, with a multivariate normal distribution that depends on $\tau(-1)$ and I_n, the identity matrix. Genes were identified from SNP coordinates using the BDGP R54/dm3 genome build. An SNP was assigned to a gene if it was +/- 1 kb from a gene body.

## Gene set enrichment analysis

GSEA was run to generate a rank-list of genes based on their enrichment for significantly associated polymorphisms as previously described (*Palu et al., 2019*). Polymorphisms within 1 kb of more than one gene were assigned to one gene based on a priority list of exon, UTR, intron, and upstream or downstream. Genes were assigned to GO categories, and calculation of enrichment score was performed as described (*Subramanian et al., 2005*). Only gene sets with $\geq$5 genes, >0.25 enrichment score, and a p<0.05 were considered.

## Evolutionary rate covariation

ERC is a method to examine the similarity of evolutionary histories of pairs of genes (*Clark et al., 2012*). The method examines the variation over time of a gene's rate of sequence evolution. Using estimates of evolutionary rate over the branches of a gene's phylogenetic tree, the method measures the correlation between genes of these branch-specific rates. Genes within correlated rate variation tend to be functionally related and have been used to discover new genes within pathways and diseases (*Brunette et al., 2019*; *Priedigkeit et al., 2015*; *Raza et al., 2019*).

ERC values in this study were taken from a compilation of ERC correlations calculated separately for three taxonomic groups: 62 mammals, 39 non-mammalian vertebrates, and 22 *Drosophila* species. Mammal and non-mammalian vertebrate alignments were taken from the multiz alignment available from the UCSC Genome Browser (*Haeussler et al., 2019*). For each alignment, we filtered out low-quality orthologs containing fewer than 50 non-gap amino acid sites or less than 70% non-gap sites and removed alignments with fewer than 15 species. Alignments were made for the *Drosophila* species after downloading protein-coding genome sequences from FlyBase and NCBI. Orthologous groups were identified using Orthofinder and alignments made with PRANK (*Emms and Kelly, 2015*; *Löytynoja and Goldman, 2008*). For each amino acid alignment, we estimated branch lengths using aaml in the phylogenetic analysis using maximum likelihood (PAML)

package (*Yang, 2007*). ERC values (correlation coefficients) for all genes with *NGLY1* were calculated using the RERconverge package (*Kowalczyk et al., 2019*). We report the ERC results for the mammalian group as the negative log of their p-values for each gene pair (*Supplementary file 5* 'nlogpvbest1'). Each gene pair also incorporated results from the vertebrate and *Drosophila* datasets by summing their negative log p-values, when orthologs were present for their respective datasets (*Supplementary file 5* 'sumnlogpvbest'). The resulting taxonomically integrated results of ERC with *NGLY1* were sorted and used for gene set enrichment analysis (GSEA).

### *Drosophila* circadian rhythm assay

Male flies with the following genotypes were used in circadian rhythm assays: *w/Y;Pdf*-GAL4/+, *yv/Y;* UAS-*Pngl*^RNAi^/+, and *yv/Y; Pdf*-GAL4/UAS-*Pngl*^RNAi^/+. Two- to 5-day-old flies were entrained for at least 3 days to a 12 hr light: 12 hr dark regimen (LD) within a *Drosophila* Activity Monitor (DAM; Tri-Kinetics, Waltham, MA) filled with standard fly food. After entrainment, flies were monitored in complete darkness (DD) for 8 days. The data was collected in 30 min bins, and analyzed for period length using ClockLab, Version 6. Graphs were generated and one-way ANOVA performed, with Tukey's multiple comparison of all three genotypes, using GraphPad Prism, Version 8.

### *Drosophila* seizure assay

The Bang Sensitivity Assay (BSA) was performed on the following genotypes: UAS-*Pngl*^RNAi^; *repo*-GAL4, UAS-*Ncc69*^RNAi^; *repo*-GAL4, and UAS-*Pngl*^RNAi^/+; UAS-*Ncc69*^RNAi^/*repo*-GAL4. Females 4–7 days old were assayed. Flies were not exposed to $CO_2$ for 3 days prior to BSA testing. Flies were flipped into empty vials and allowed to rest for 2 hr. They were then vortexed on a Thermo Scientific LP Vortex Mixer for 10 s at maximum speed. The vortexed flies were filmed for 60 s. The video was used to score seizures at 5, 10, 30, and 60 s.

## Mammalian cell culture and proteasome inhibition

MEFs were generated by Jackson Laboratory (Bar Harbor, Maine) from *NGLY1* knockout mice and littermate controls (C57BL/6J-*Ngly1*^em4Lutzy^/J, #027060). MEFs were immortalized in the laboratory of Dr. Hamed Jafar-Nejad (Baylor College of Medicine) and then gifted to us. MEFs were grown in DMEM (Gibco 11965) supplemented with 10% fetal bovine serum (FBS) and penicillin/streptomycin in 5% $CO_2$ at 37°C. For proteasome inhibition, MEFs were incubated with 500 nM bortezomib (EMD Millipore) or an equivalent volume of DMSO as a vehicle control, for 4 hr under standard conditions.

## Western blotting

MEFs were grown to 80–90% confluency then collected. Cell pellets were weighed and then resuspended in a proportional volume of phosphate buffered saline (PBS). Equivalent volumes of resuspension were always used for each lysis. Cells were lysed using a cell fractionation kit (Cell Signaling Technologies, #9038) with each buffer supplemented with 1 mM PMSF and 1x protease inhibitor cocktail (Cell Signaling Technologies, #5871).

Lysates were separated by SDS-PAGE on 3–8% Tris-acetate gels (BioRad #3450129) for 2.5 hr at 150V, then transferred to PVDF membrane by wet transfer at 50V for 1 hr. Membranes were blocked in either 5% milk or 5% BSA according to the recommendations of the primary antibody manufacturer. Primary antibodies were as follows: anti-NKCC1 (Cell Signaling Technologies #14581), anti-TCF11/NRF1 (Cell Signaling Technologies #8052). Membranes were incubated in primary antibody at 1:1000 in blocking buffer overnight. IRDye secondary antibody (Abcam #216773) was used for infrared detection at 1:10,000 dilution in blocking buffer for 1 hr. Membranes were scanned on an Odyssey CLx (Li-cor) and analyzed with the accompanying software, Image Studio.

## Deglycosylation reactions

MEFs were lysed in the same manner as described for western blotting. The membrane fraction was then incubated with one of the three deglycosylation enzymes: O-Glycosidase (New England Biolabs, #P0733), PNGase F (New England Biolabs, #P0704), Endoglycosidase F (New England Biolabs, #P0702) according to the manufacturer's directions. Reactions were incubated at 37°C for 1 hr. Controls were treated with all the same buffers and reaction conditions but without the added enzyme.

## Rb$^+$ flux assay

20,000 cells/well of immortalized MEFs from control or *Ngly1 -/-* mice were seeded into a 96-well plate. The following day, media was removed and the cells were washed with 1x PBS. Of pre-incubation medium (in mM, 135 Na gluconate, 5 K gluconate, 1 Ca gluconate, 1 Mg gluconate, 15 HEPES pH 7.4, 5 glucose), 100 µl was added to each well and the cells were incubated for 30 min at 37°C. Next, 100 µl of pre-incubation medium containing either DMSO, bumetanide, or ouabain was added to each well to achieve final concentrations of 0.1% (DMSO), 10 µM (bumetanide), or 0.1 mM (ouabain) and incubated for 30 min at room temperature. Then, 150 µL of medium containing DMSO (0.1%), bumetanide (10 µM) or ouabain (0.1 mM), as well as $^{86}$Rb (10 mCi/µl), was added to each well. Three different media were used. Isotonic media contained (in mM): 140 NaCl, 5 KCl, 2 CaCl$_2$, 1 MgCl$_2$, 5 glucose, 15 HEPES pH 7.4. Hypertonic medium was the same as isotonic medium, with the addition of 75 mM sucrose. For hypotonic medium, isotonic medium was diluted 1:2 in water. The cells were incubated for 7 min at room temperature. Medium was removed and the cells were washed three times with ice-cold 1x PBS. Cells were lysed in 100 µl 2% SDS and incubated for 15 min at room temperature. Radioactivity was measured in a liquid scintillation counter.

## Acknowledgements

This study is dedicated to Bertrand Might, the first child diagnosed with NGLY1 deficiency. We thank Dr. Hamed Jafar-Nejad (Baylor College of Medicine) for the gift of immortalized *NGLY1*-null MEFs. This research was supported by the NIH through an NIGMS R35 award (R35GM124780) (CYC), NIDDK R01 award (R01 DK110358) (ARR), and NHGRI R01 award (R01 HG009299) (NLC). This work was also supported by a Glenn Award from the Glenn Foundation for Medical Research to CYC. CYC was the Mario R Capecchi Endowed Chair in Genetics. DMT and KSH were supported on an NIH/NHGRI Genomic Medicine T32 postdoctoral training grant from the University of Utah (T32 HG008962) and by a generous gift from the Might family through the Bertrand T Might Fellowship. KGO was supported by the NIH/NIGMS Genetics T32 Fellowship from the University of Utah (T32 GM007464). The MEFs were derived from NGLY1 knockout mice which were funded by the Grace Science Foundation to CML.

## Additional information

### Funding

| Funder | Grant reference number | Author |
| --- | --- | --- |
| National Institute of General Medical Sciences | R35GM124780 | Clement Y Chow |
| National Institute of Diabetes and Digestive and Kidney Diseases | R01DK110358 | Aylin R Rodan |
| National Human Genome Research Institute | R01HG009299 | Nathan L Clark |
| Glenn Foundation for Medical Research | Glenn Award | Clement Y Chow |
| National Human Genome Research Institute | T32HG008962 | Dana M Talsness Kevin A Hope |
| Might family | Bertrand T Might Fellowship | Dana M Talsness |
| National Institute of General Medical Sciences | T32GM007464 | Katie G Owings |

The funders had no role in study design, data collection and interpretation, or the decision to submit the work for publication.

## Author contributions
Dana M Talsness, Data curation, Formal analysis, Investigation, Visualization, Writing - original draft, Writing - review and editing; Katie G Owings, Data curation, Formal analysis, Visualization; Emily Coelho, Data curation, Formal analysis; Gaelle Mercenne, John M Pleinis, Kevin A Hope, Data curation; Raghavendran Partha, Formal analysis; Aamir R Zuberi, Cathleen M Lutz, Resources; Nathan L Clark, Formal analysis, Supervision, Writing - review and editing; Aylin R Rodan, Supervision, Writing - review and editing; Clement Y Chow, Conceptualization, Formal analysis, Supervision, Funding acquisition, Writing - original draft, Writing - review and editing

## Author ORCIDs
Dana M Talsness (iD) https://orcid.org/0000-0001-7823-1616
Raghavendran Partha (iD) http://orcid.org/0000-0002-7900-4375
Aylin R Rodan (iD) http://orcid.org/0000-0001-9202-2378
Clement Y Chow (iD) https://orcid.org/0000-0002-3104-7923

## Decision letter and Author response
Decision letter https://doi.org/10.7554/eLife.57831.sa1
Author response https://doi.org/10.7554/eLife.57831.sa2

# Additional files

### Supplementary files
• Supplementary file 1. *NGLY1* DGRP cross progeny counts. The number of eclosed flies were scored for each resulting genotype. The 'no marker' column represents flies expressing the NGLY1 RNAi. The largest balanced genotype was used as 'expected' for percent survival.

• Supplementary file 2. GWA analysis for survival in *NGLY1* DGRP screen. Single-nucleotide polymorphisms (SNPs) are listed by chromosome position and rs ID.

• Supplementary file 3. Top associated SNPs. The top 125 variants. SNPs are listed in rank order of significance.

• Supplementary file 4. Gene set enrichment analysis (GSEA). Gene Ontology (GO) terms are listed by rank significance. Individual genes within each category are listed with the FBgn#.

• Supplementary file 5. Evolutionary rate covariance (ERC). Co-evolving genes are listed by rank significance (sumnlogpvbest). Genes that are known to cause a Congenital Disorder of Glycosylation (CDG) are highlighted in red.

• Transparent reporting form

### Data availability
All data generated by this study are included in the manuscript and supporting files.

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
