## [Decision Letter]

[Editors’ note: the authors submitted for reconsideration following the decision after peer review. What follows is the decision letter after the first round of review.]

Thank you for choosing to send your work, "A *Drosophila* screen identifies NKCC1 as a substrate of NGLY1 deglycosylation and a modifier of NGLY1 deficiency", for consideration at *eLife*. Your article has been reviewed by two peer reviewers, and the evaluation has been overseen by a Reviewing Editor and a Senior Editor. The reviewers have opted to remain anonymous. Although the work is of interest, we regret to inform you that the findings at this stage are too preliminary for further consideration at *eLife*.

The reviewers are all positive about your screen and clearly appreciate your work. However they question the mechanism that you propose. I suggest that you carefully read their critiques. As you will notice they are all three experts in the glycosylation field and make some excellent suggestions. If you think you can address their concerns, please write a rebuttal letter and I will share it with the reviewers. We can then determine a mode of action.

Reviewer #1:

In manuscript Talsness and colleagues use *Drosophila* genetics combined with evolutionary rate covariation analysis and experiments in mouse embryonic fibroblasts (MEFs) to identify the ion transporter NKCC1 as a genetic modifier and potentially a direct target of NGLY1. The authors started by an elegant genetic screen to ask whether any of the natural variations found in 163 genetically heterogeneous fly strains from the DGRP collection can suppress the lethality associated with ubiquitous NGLY1 knock-down. The screen helped them identify a number of DGRP strains that can reduce the lethality of NGLY1-KD in a rather wide range, some to astonishingly high levels (up to ~97% survival). They next performed genome-wide association studies (GWAS) to identify variants in these strains that show statistically significant association with the survival of NLGY1-KD animals. This resulted in the identification of variants in or close to a number of genes, some of which are related to pathways like ERAD and proteasomal degradation to which NGLY1 has previously been linked. As proof of principle, they chose one category (circadian rhythm) and performed tissue-specific RNAi to show that indeed NGLY1-KD in brain pacemaker neurons results in altered circadian rhythm in flies. As a complementary approach to the above screen, the authors performed evolutionary rate covariation (ERC) analysis and identified several interesting GO categories that seem to co-evolve with NGLY1, suggesting potential functional links. Moreover, two of the top ranking GWAS candidates from the genetic screen were also found in the ERC analysis.

The authors then chose the top ranking GWAS candidate with a human homolog for functional studies. This gene is called Ncc69 in flies and encodes a glycoprotein homologous to mammalian cation-chloride co-transporters NKCC1 and NKCC2. Based on two sets of double-KD experiments, the authors concluded that Ncc69 genetically interacts with NGLY1 in flies. They then performed Western blots on NGLY1-mutant and control MEFs with or without glycosidase treatment or proteasome inhibitor treatment, and concluded based on the data that NKCC1 (the homolog expressed in MEFs) is a substrate for deglycosylation by NGLY1. Finally, they used the transport of radioactive rubidium as a functional readout for NKCC1 activity and provided evidence suggesting that the NKCC1 activity is reduced ~50% in NGLY1-mutant MEFs. Based on all these data, the authors concluded that deglycosylation of NKCC1 by NGLY1 is required for NKCC1's full activity. Finally, the authors discussed the potential new avenues that their work can open in the NGLY1 field, the possibility that some NGLY1 deficiency phenotypes like lack of tears are due to reduced NKCC1 function, and that this ion transporter might be a novel therapeutic target for this disease.

The genetic screen and the follow-up GWAS analysis reported in this manuscript provides a beautiful example of the broad and sometime extreme effects of genetic background on a developmental phenotype and the possibility of identifying novel potential modifiers of a disease gene phenotype in an unbiased manner by these techniques. This also seems to be the first report on the circadian rhythm abnormalities in an animal model of NGLY1 deficiency, with potential mechanistic link to the sleep abnormalities that are commonly seen in the affected patients. Moreover, the GWAS and ERC datasets presented in this work are likely to be help push the research in this field forward. Unfortunately, there are a number of issues with the experiments performed on Ncc69/NKCC1, which in my opinion cast doubt on the conclusions of the manuscript. This is in part due to lack of sufficient controls in some experiments.

Genetic interaction between NGLY1 and Ncc69 in flies

The authors have used a double KD strategy to examine the genetic interaction between NGLY1 and Ncc69. They state that NGLY1 KD with tubulin-Gal4 did not cause significant lethality (Figure 3A). Wasn't the same genotype used in the genetic screen and mentioned to show lethality, prompting the authors to use a tubulin-Gal80 in the background? The authors' previous work (Owings et al) also indicated only ~30% eclosion to adulthood in tubulin-Gal4 UAS-Ngly1-RNAi. The authors should address the reason for this discrepancy and repeat the genetic interaction studies in the same context as the original screen. Moreover, given the potential off target effects with RNAi, the genetic interaction should be confirmed by either rescue by an RNAi-resistant transgene of one of these genes (for example, the human homolog), or by using loss-of-function allele(s) (or at least by using additional independent RNAi strains for each gene). If I understood correctly, the authors have found Ncc69 SNPs in 9 of the DGRP strains which modified the NGLY1 KD lethality. How are these 9 strains distributed across the data shown in Figure 1B? Based on the statement ("that deletions encompassing these nine Ncc69 intronic SNPs result in a null allele") and in light of the genetic interaction suggested in Figure 3A, one would think that Ncc69-lines from DGRP would show a high degree of lethality when combined with NGLY1-KD. Do the authors find it surprising that so many SNPs in this gene were actually associated with survival to adulthood in NGLY1 KD animals? This might be caused by an increase in Ncc69 expression in those strains and should be tested. In fact, if what I have written above is correct, it would have been better to show that Ncc69 overexpression can rescue the lethality of NGLY1-KD animals.

The data in Figure 3B seems to be in the same direction as the results of the genetic screen. However, it has the same technical issues as 3A, in the sense that only one RNAi has been used for each gene and the degree of KD is not known. In this case, it will be more difficult to determine KD efficiency, so using independent RNAi lines and/or heterozygosity for NGLY1 in an Ncc69-KD background might be good alternatives. For 3B, please show a control or state the percentage of the control animals that show bang sensitivity.

NKCC1 as a direct target of NGLY1

There is discrepancy between the data shown in Figure 4A and 4D. In 4A, the NKCC1 bands show a bigger apparent size in Ngly1-/- cells compared to control cells, compatible with the idea that NGLY1 normally deglycosylates some NKCC1 molecules. However, in control lanes of 4D for all three digestions, the NKCC1 bands in Ngly1-/- and control cells look similar to each other. It is hard to explain these data, and they make the digestion results inconclusive. The idea is that in Ngly1-/- cells, NKCC1 becomes bigger (N-glycan retention) and then upon PNGase F and potentially Endo-H digestion it loses N-glycans and becomes the same size as the NKCC1 from wild-type cells. But if the two cells start with the same pattern for a potential target protein, then the glucosidase digestion results apply to those N-glycans that would not normally be removed by NGLY1 (i.e., N-glycans on properly folded NKCC1 molecules) and therefore cannot be used to conclude that the protein is normally deglycosylated by NGLY1. In addition, given that ERAD substrates do not traffic to Golgi, they should harbor high mannose glycans. Therefore, when an NGLY1 target is not deglycosylated in Ngly1-/- cells, the retained glycans should be Endo-H sensitive, not Endo-H resistant.

On a related note, the authors wrote "Together these data indicate that NGLY1 has deglycosylation activity that is independent of both the retrotranslocation and the degradation components of the ERAD pathway." This conclusion is against everything so far discovered about NGLY1. The authors are cognizant of this issue and hypothesize in the paragraph following this sentence that an asparagine in the cytoplasmic region of NKCC1 might be N-glycosylated by an unknown enzymatic machinery and deglycosylated by NGLY1. They refer to a dog kidney sodium pump as an example of a protein with a cytoplasmic N-glycosylation. However, to my knowledge, presence of N-glycans on this protein or any other cytoplasmic protein has not been confirmed by mass spectrometry and site-directed mutagenesis (please provide references to relevant research articles if there are such reports). Moreover, the identity of the cytoplasmic enzymes that would presumably add N-glycans to cytoplasmic proteins remains unknown. This is not to say that we should not consider any idea that is outside of what has been proposed about N-glycosylation and NGLY1 function. However, to make this argument, the authors need to at least make a mutation in that site (N168) and test whether NKCC1-N184Q shows a decrease in apparent molecular weight compared to NKCC1-WT in control cells, fails to shift up in Western blots in Ngly1-/- cells, and ideally whether NKCC1-N184Q can rescue the effect of NGLY1 loss on the function of NKCC1. Similar experiments on the two confirmed N-glycosylation sites of this protein would make the authors' conclusion about NKCC1 being a target of NGLY1 much stronger.

Reviewer #2:

The manuscript presents a powerful approach for identifying modifying genes that may be relevant to a human genetic disease whose clinical presentation is very heterogeneous. The background, results, and data discussion are well organized and clearly written. In general, the conclusions are supported by the data. Although additional discussion and consideration of alternative hypotheses could be expanded. Substantive concerns are as follows:

1) Tadashi Suzuki's group identified and published the importance of genetic background for NGLY1 deficiency in mouse in 2017. The authors cite this work (Fujihara, et al.) in their Introduction, but only in relation to the identification of endogenous NGLY1 substrates by mass spectrometry. Additional credit and discussion should be given to the Suzuki observations related to the impact of genetic background, a major aspect of this current manuscript.

2) The authors pursue orthogonal validation and further characterization of one of their top modifier hits, NKCC1. This work is well done, includes appropriate controls, and is described clearly. However, the authors provide little insight into how NKCC1, a cell surface protein modified with complex glycans (according to the authors), is able to come into contact with NGLY1. The authors assert that it must because the mass of the protein revealed by SDS-PAGE and western blot is larger in the KO MEFs than in control MEFs. The authors initially propose that NKCC1 carries two glycans in the KO and only 1 in the control. Based on the topology model for the protein and the canonical sequence (uniprot accession P55012-1) two N-liked sequons sit very close to each other in an extracellular loop and three other sequons are predicted to be on cytoplasm-oriented loops. The sequon information is summarized at GlyGen Protein Detail page:

https://www.glygen.org/glycoprotein_detail.html?uniprot_canonical_ac=P55012-1&listID=8dd9dc26a96c1912593c86492827bbb7&gs=NKCC1

The authors recognize a sequon at N163, predicted to be cytoplasm oriented and suggest that a cytoplasmic glycosylation event may occur by an unidentified enzyme or mechanism. Glycosylation at this site would seem to bring the total to 3 sites, not 2. They do not mention the other cytoplasmic glycosylation sequons at N599 and N683. These Asn residues would also seem to be reasonable candidates for cytoplasmic N-glycosylation if such a process exists. The problem that the authors need to more effectively discuss is that no such process is known. In order for a complex glycan to be placed on a cytoplasmic sequon, not only would the unknown enzyme have to transfer a precursor from a Dol-P-donor (localized to the ER lumen), but that modification would then have to be accessible to the entire secretory pathway to be processed toward EndoH resistance. How do the authors envision this happening?

The only definitive way to test this hypothesis is to obtain proteomic characterization of the protein expressed in wild type and KO MEFS. If any of the cytoplasmic sequon Asn residues can be shown to be converted to Asp residues in wild type but not in KO, the authors would have very strong support for their hypothesis. This sort of experiment falls into the category of perhaps beyond the resources or expertise of the authors and is probably an undertaking that would take significant effort. Nonetheless, it should be acknowledged somewhere in the Discussion that this sort of information would provide definitive support for the mechanism they propose.

In the absence of this proof, the proposal that a cytoplasmic glycosylation machinery may exist seems less likely than other possibilities that could be discussed. For instance, the authors do offer the possibility that NGLY1 has functions independent of its enzymatic activity. Perhaps one of those activities influences the efficiency of the oligosaccharyltransferase complex such that it is less likely to modify both of the extracellular Asn residues. Loss of NGLY1 might then lead to more efficient glycosylation of NKCC1. This mechanism does not require the invocation of an unknown cytoplasmic N-linked glycosylation machinery. A more well-reasoned discussion of other possibilities is warranted.

Reviewer #3:

In this manuscript the authors examined how genetic modifiers affect the phenotypes of flies where Pngl (fly homolog of NGLY1) is knocked down by RNAi. The authors crossed the Pngl knockdown flies with a collection of genetically diverse *Drosophila* strains (*Drosophila* Genetic Reference Panel). Using GWA analysis, they identified 125 variants that associated with enhanced survival. After detailed analysis of their results (GSEA, ERC, GO), they chose to analyze one of the top hits, Ncc69, the fly homolog of the conserved ion transporters, NKCC1 and 2. They confirmed a genetic interaction between Pngl and Ncc69 by knocking both down, which resulted in synthetic lethality. They also knocked down Ncc69 alone in glial cells, which showed a "bang" seizure phenotype, which was partially rescued in the Pngl knockdown flies. Although the phenotypes were not consistent (rescue in the initial screen, synthetic lethality, rescue of seizure phenotype), these data suggested some sort of genetic interaction. To see if this held true in mammals, they analyzed NKCC1 in WT and NGLY1-null mouse embryonic fibroblasts (MEFs). Western blots showed NKCC1 migrating as a large smear in the WT cells, centering around 150 kDa, but the lower half of the smear was less intense in the NGLY-1 samples. PNGaseF digestions of both samples resulted in a band migrating at the same place, suggesting that the difference involved N-glycans. Lack of digestion with EndoH suggested the difference was in complex-type N-glycans. They also performed ion flux assays and showed that the NGLY1-null cells had a lower Bumetanide-sensitive ion flux (due to NKCC1) than the WT cells. Based on these data the authors concluded the NKCC1 is a direct substrate for NGLY1 de-glycosylation and that this de-glycosylation alters NKCC1 function.

1) Although most of this work is well done, the data does not support the conclusion that NKCC1 is a direct substrate for NGLY1. As the authors point out, the two predicted N-glycans are on an extracelluar loop of NKCC1 and are of the complex-type, so it is unlikely that a cytoplasmically localized NGLY1 could access the sites to de-glycosylate them. Instead, the authors invoked a cytoplasmic N-glycosylation event with an unknown cytoplasmic glycosyltransferase modifying N168, which is cytoplasmic. The authors reference a review by Hart and Wells, 2017, which describes a report that the α-subunit of the dog kidney sodium pump Na+, K^+^-ATPase is modified on a cytoplasmic domain with an N-glycan. The paper referred to was published in 1990 (https://www.ncbi.nlm.nih.gov/pubmed/2175915 ) and has not been rigorously confirmed using mutagenesis or mass spectral glycoproteomic site mapping. In the review, Hart and Wells conclude, "This provocative claim has remained unresolved". In addition, the large shift in size in Figure 4A, estimated to be as much as 20 kDa (Figure 4C) suggests that the change is much more than a simple complex-type N-glycan. A typical biantennary complex N-glycan with two sialic acids has a mass of just over 2 kDa. This suggests that the N-glycans on the extracellular loop of NKCC1 are extended in some way, possibly by poly-N-acetyl-lactosamine repeats or polysialic acid. Thus, a much simpler explanation for the change in molecular weight in Figure 4A is that loss of NGLY1 in these cells induces, indirectly, expression of glycosyltransferases responsible for enhanced extension of the 2 N-glycans on the extracellular loop of NKCC1. Without more experimentation, the authors cannot conclude that NKCC1 is a direct substrate of NGLY1. Other possibilities are more likely.

2) It is not clear why KD of Ncc69 rescued the Pngl phenotype in the initial screen while KD of both in Figure 3A caused synthetic lethality.

[Editors’ note: further revisions were suggested prior to acceptance, as described below.]

Thank you for submitting your article "A *Drosophila* screen identifies NKCC1 as a modifier of NGLY1 deficiency" for consideration by *eLife*. Your article has been reviewed by two peer reviewers, and the evaluation has been overseen by Reviewing Editor Hugo Bellen and Patricia Wittkopp, Senior Editor. The reviewers have opted to remain anonymous.

We are glad to conditionally accept the manuscript for publication in *eLife*. The reviewers have discussed the reviews with one another and the Reviewing Editor has drafted this decision to help you prepare a revised submission.

In the revised version, the authors have addressed my comments, in some cases by additional experiments but more often by changing the text to tone down the previous conclusions. The revised manuscript is improved, as most of the statements made in the manuscript are supported by the data. The mechanistic link between NGLY1 and NKCC1 remains largely unknown, and this makes several paragraphs in the Discussion section rather speculative. However, the fly and bioinformatic screens, the genetic interactions, and the reduced NKCC1 function in Ngly1 mutant MEF cells, combined with the phenotypic similarities between NKCC1 and NGLY1 mutants in humans and mice, are important contributions to our understanding of NGLY1 biology and potentially the pathophysiology of NGLY1 deficiency.

Please address the following points:

Impact statement: Please add "potentially" before "explaining several symptoms of NGLY1 deficiency such as lack of sweat and tears.".

The authors write "there is no evidence of ER stress upon RNAi knockdown of NGLY1 in *Drosophila*, (Owings et al., 2018) or upon loss of NGLY1 function in mouse, rat, and human cells (Asahina et al., 2020; Mueller et al., 2020; Tambe, Ng and Freeze, 2019)." Also, in the Discussion a similar statement has been made. Loss of Ngly1 in mouse embryonic fibroblasts has recently been shown to significantly increase the level of several ER stress markers (BiP, phopho-IRE1α and OS9), and this was further enhanced when an NGLY1 substrate was overexpressed in these cells (Galeone et al., 2020). This paper has been already cited by the authors in a different part of the manuscript. Please revise these sentence to incorporate the findings of this report.

In the response letter, the authors have acknowledged that the PNGase F and Endo H data are difficult to interpret and have agreed with the reviewers' comments that the data are not sufficient for calling NKCC1 a target of NGLY1. In accordance with this, the following sentence (between quotation marks) should be removed, as it implies otherwise (The authors wrote: In cells derived from an NGLY1 -/- mouse model we found that NKCC1 protein migrated at a higher molecular weight relative to +/+ cells. "Treatment with PNGase eliminated this size difference, confirming that this is due to N-linked glycosylation.")

A similar sentence that needs to be removed, as it is not supported by the data: "but rather NKCC1 has a molecular weight shift that can be eliminated with PNGase treatment".

Just for the authors' information: In response to question #3 of reviewer 3, the authors mentioned that they have used attP2 and attP40 lines without RNAi insertion as negative control. Sometimes adding a second UAS transgene (with or without RNAi) can dilute the effects of a GAL4 driver on a specific RNAi line. In other words, to prove that the partial rescue of the Ncc69 KD phenotype by adding UAS-Ngly1-RNAi chromosome is due to simultaneous Ngly1 KD (shown in Figure 3B), it would have been better to use a UAS-GFP line (to generate UAS-GFP; UAS-Ncc69-RNAi; repo-GAL4 animals) to show that it's the Ngly1 KD that is partially rescuing, not the UAS element. This only becomes important when the second transgene improves the phenotype of the first transgene (like 3B) and is not necessary when the phenotype is enhanced (like 3A).

---

## [Author Response]

[Editors’ note: The authors appealed the original decision. What follows is the authors’ response to the first round of review.]

Reviewer #1:[…]Genetic interaction between NGLY1 and Ncc69 in fliesThe authors have used a double KD strategy to examine the genetic interaction between NGLY1 and Ncc69. They state that NGLY1 KD with tubulin-Gal4 did not cause significant lethality (Figure 3A). Wasn't the same genotype used in the genetic screen and mentioned to show lethality, prompting the authors to use a tubulin-Gal80 in the background? The authors' previous work (Owings et al) also indicated only ~30% eclosion to adulthood in tubulin-Gal4 UAS-Ngly1-RNAi. The authors should address the reason for this discrepancy and repeat the genetic interaction studies in the same context as the original screen.

It is true that this same NGLY1 RNAi line showed ~30% eclosion in our previous publication (Owings et al., 2018). In the process of preparing this manuscript, we were also surprised by the significant change in lethality for this line (multiple experiments were consistent). After receiving the reviews, we suspected contamination of the original NGLY1 RNAi line, to correct this we re-ordered the strains and repeated these crosses. Our new results show ~25% eclosion, similar to our previous publication. We have replaced Figure 3A and the wording in the Results section.

It should be noted that the potential contamination does affect the screen data, as that was performed in 2016 and the follow-up experiments in question were performed in 2019.

The data in Figure 3B seems to be in the same direction as the results of the genetic screen. However, it has the same technical issues as 3A, in the sense that only one RNAi has been used for each gene and the degree of KD is not known. In this case, it will be more difficult to determine KD efficiency, so using independent RNAi lines and/or heterozygosity for NGLY1 in an Ncc69-KD background might be good alternatives.

The *NGLY1* RNAi was previously validated (Owings et al., 2018). In that publication we used multiple *NGLY1* RNAi lines which showed similar results. We have added this explanation into the Results section of the manuscript. The *Ncc69* RNAi line was validated by another group (Leiserson, Forbush and Keshishian, 2011) and we have added this reference to the manuscript as well. As far as using heterozygous *NGLY1*-null animals, we have found that they do not have seizures.

For 3B, please show a control or state the percentage of the control animals that show bang sensitivity.

We apologize for not adding in this data. The percentage of control animals that show bang sensitivity is 0%, and we have added this control group into Figure 3 and in the text.

Moreover, given the potential off target effects with RNAi, the genetic interaction should be confirmed by either rescue by an RNAi-resistant transgene of one of these genes (for example, the human homolog), or by using loss-of-function allele(s) (or at least by using additional independent RNAi strains for each gene). If I understood correctly, the authors have found Ncc69 SNPs in 9 of the DGRP strains which modified the NGLY1 KD lethality. How are these 9 strains distributed across the data shown in Figure 1B?

We apologize for not being clear. There were not 9 DGRP strains that contained SNPs, rather there were many strains that contained a number of these 9 SNPs. These 9 SNPs are not independent, rather, they are contained in a 54 bp region and are in near perfect linkage disequilibrium. We have revised this section so that it is more clear.

Based on the statement ("that deletions encompassing these nine Ncc69 intronic SNPs result in a null allele") and in light of the genetic interaction suggested in Figure 3A, one would think that Ncc69-lines from DGRP would show a high degree of lethality when combined with NGLY1-KD. Do the authors find it surprising that so many SNPs in this gene were actually associated with survival to adulthood in NGLY1 KD animals? This might be caused by an increase in Ncc69 expression in those strains and should be tested. In fact, if what I have written above is correct, it would have been better to show that Ncc69 overexpression can rescue the lethality of NGLY1-KD animals.

We apologize for being very unclear in this section. These 9 linked SNPs are in a region encompassed by a much larger deletion allele of the *Ncc69* gene. This statement was imprecise and was simply meant to convey that this region was deleted as part of a larger deletion and not meant to convey that these 9 SNPs are causative of the deleted allele. We eliminated this statement to prevent further confusion. Additionally, we have analyzed *Ncc69* expression levels in the DGRP strains using previously published RNAseq data from the 200 strains. We found that there was no correlation between Ncc69 expression and survival and have included the data in the text and as Figure 1—figure supplement 1. This lack of correlation is not particularly surprising, as the expression data available is from whole, adult flies and we are not sure which time points or tissues are most important for the genetic interaction we identified.

NKCC1 as a direct target of NGLY1There is discrepancy between the data shown in Figure 4A and 4D. In 4A, the NKCC1 bands show a bigger apparent size in Ngly1-/- cells compared to control cells, compatible with the idea that NGLY1 normally deglycosylates some NKCC1 molecules. However, in control lanes of 4D for all three digestions, the NKCC1 bands in Ngly1-/- and control cells look similar to each other. It is hard to explain these data, and they make the digestion results inconclusive. The idea is that in Ngly1-/- cells, NKCC1 becomes bigger (N-glycan retention) and then upon PNGase F and potentially Endo-H digestion it loses N-glycans and becomes the same size as the NKCC1 from wild-type cells. But if the two cells start with the same pattern for a potential target protein, then the glucosidase digestion results apply to those N-glycans that would not normally be removed by NGLY1 (i.e., N-glycans on properly folded NKCC1 molecules) and therefore cannot be used to conclude that the protein is normally deglycosylated by NGLY1.

We agree that the controls for the enzyme digestion make it difficult to interpret the results. The size difference observed in Figure 4A is very reproducible (also see Figure 4E). We have since repeated this experiment in Figure 4D multiple times with very similar results. We believe that incubating the lysate at 37°C for 60 minutes decreases the integrity of the 150kDa, 12-pass membrane, glycoprotein NKCC1 and therefore it is difficult to see the size difference in the untreated state. Because the difference is not an “all or none” difference in Figure 4A, we think that this instability makes it even harder to observe in the control treatments. However, we believe that in our most recent Endo H treatment the size difference is more visible and therefore we have replaced this blot in Figure 4D.

In addition, given that ERAD substrates do not traffic to Golgi, they should harbor high mannose glycans. Therefore, when an NGLY1 target is not deglycosylated in Ngly1-/- cells, the retained glycans should be Endo-H sensitive, not Endo-H resistant.

We regret that our language made it sounds as if NKCC1 was an ERAD substrate. We believe that by eliminating all of the language about NKCC1 being a direct substrate of NGLY1 deglycosylation, that we have also removed the assumption that it is undergoing ERAD, and therefore it is completely plausible that they should be Endo-H resistant.

On a related note, the authors wrote "Together these data indicate that NGLY1 has deglycosylation activity that is independent of both the retrotranslocation and the degradation components of the ERAD pathway." This conclusion is against everything so far discovered about NGLY1. The authors are cognizant of this issue and hypothesize in the paragraph following this sentence that an asparagine in the cytoplasmic region of NKCC1 might be N-glycosyalted by an unknown enzymatic machinery and deglycosylated by NGLY1. They refer to a dog kidney sodium pump as an example of a protein with a cytoplasmic N-glycosylation. However, to my knowledge, presence of N-glycans on this protein or any other cytoplasmic protein has not been confirmed by mass spectrometry and site-directed mutagenesis (please provide references to relevant research articles if there are such reports). Moreover, the identity of the cytoplasmic enzymes that would presumably add N-glycans to cytoplasmic proteins remains unknown. This is not to say that we should not consider any idea that is outside of what has been proposed about N-glycosylation and NGLY1 function. However, to make this argument, the authors need to at least make a mutation in that site (N168) and test whether NKCC1-N184Q shows a decrease in apparent molecular weight compared to NKCC1-WT in control cells, fails to shift up in Western blots in Ngly1-/- cells, and ideally whether NKCC1-N184Q can rescue the effect of NGLY1 loss on the function of NKCC1. Similar experiments on the two confirmed N-glycosylation sites of this protein would make the authors' conclusion about NKCC1 being a target of NGLY1 much stronger.

We completely agree that site directed mutagenesis is the next step in this scientific story, and have noted such in the Discussion section. However, as we stated in our rebuttal letter, we believe that these experiments are beyond the scope of this manuscript.

Reviewer #2:The manuscript presents a powerful approach for identifying modifying genes that may be relevant to a human genetic disease whose clinical presentation is very heterogeneous. The background, results, and data discussion are well organized and clearly written. In general, the conclusions are supported by the data. Although additional discussion and consideration of alternative hypotheses could be expanded. Substantive concerns are as follows:1) Tadashi Suzuki's group identified and published the importance of genetic background for NGLY1 deficiency in mouse in 2017. The authors cite this work (Fujihara, et al.) in their Introduction, but only in relation to the identification of endogenous NGLY1 substrates by mass spectrometry. Additional credit and discussion should be given to the Suzuki observations related to the impact of genetic background, a major aspect of this current manuscript.

Thank you for pointing this out. We regret this oversight and we have added the following sentence into the Introduction, “This variability based on background genetics was also observed in the lab when an NGLY1 deficiency mouse model was crossed onto an outbred mouse strain which partially rescued the lethality of the model (Fujihira et al., 2017)”.

2) The authors pursue orthogonal validation and further characterization of one of their top modifier hits, NKCC1. This work is well done, includes appropriate controls, and is described clearly. However, the authors provide little insight into how NKCC1, a cell surface protein modified with complex glycans (according to the authors), is able to come into contact with NGLY1. The authors assert that it must because the mass of the protein revealed by SDS-PAGE and western blot is larger in the KO MEFs than in control MEFs. The authors initially propose that NKCC1 carries two glycans in the KO and only 1 in the control. Based on the topology model for the protein and the canonical sequence (uniprot accession P55012-1) two N-liked sequons sit very close to each other in an extracellular loop and three other sequons are predicted to be on cytoplasm-oriented loops. The sequon information is summarized at GlyGen Protein Detail page:https://www.glygen.org/glycoprotein_detail.html?uniprot_canonical_ac=P55012-1&listID=8dd9dc26a96c1912593c86492827bbb7&gs=NKCC1The authors recognize a sequon at N163, predicted to be cytoplasm oriented and suggest that a cytoplasmic glycosylation event may occur by an unidentified enzyme or mechanism. Glycosylation at this site would seem to bring the total to 3 sites, not 2. They do not mention the other cytoplasmic glycosylation sequons at N599 and N683. These Asn residues would also seem to be reasonable candidates for cytoplasmic N-glycosylation if such a process exists.

The reviewer is correct that there are three additional N-X-S/T sites in the mouse (and human) NKCC1. We have added this sentence to the Discussion to clarify our reasoning, “Amino acid analysis reveals three other asparagine residue within the necessary N-X-S/T sequence for N-linked glycosylation, however two are predicted to be in transmembrane domains; the third (human NKCC1 residue N168) is located in the amino-terminal cytoplasmic tail of the protein.”.

The problem that the authors need to more effectively discuss is that no such process is known. In order for a complex glycan to be placed on a cytoplasmic sequon, not only would the unknown enzyme have to transfer a precursor from a Dol-P-donor (localized to the ER lumen), but that modification would then have to be accessible to the entire secretory pathway to be processed toward EndoH resistance. How do the authors envision this happening?

At this point, the exact mechanism is unclear. We apologize for jumping to conclusions that were not warranted by our data. We believe that by eliminating all of the language about NKCC1 being a direct substrate of NGLY1 that we have eliminated these concerns.

The only definitive way to test this hypothesis is to obtain proteomic characterization of the protein expressed in wild type and KO MEFS. If any of the cytoplasmic sequon Asn residues can be shown to be converted to Asp residues in wild type but not in KO, the authors would have very strong support for their hypothesis. This sort of experiment falls into the category of perhaps beyond the resources or expertise of the authors and is probably an undertaking that would take significant effort. Nonetheless, it should be acknowledged somewhere in the Discussion that this sort of information would provide definitive support for the mechanism they propose.

Since the review of our manuscript, we were able to send a gel slice corresponding to the molecular weight range of NKCC1 for mass spec analysis. The analysis was able to identify several peptides mapping to NKCC1 (SLC12A2), see Author response image 1.

However, these peptides gave only ~10% protein coverage, as was predicted by the personnel at our mass spec core given the hydrophobic nature of the 12-pass transmembrane domains. Unfortunately, none of the peptides included a potential N-linked glycosylation site. We plan to perform more mass spec analysis with varied parameters, but we believe this is beyond the scope of this manuscript. As suggested, we have added language to the Discussion about future mass spec analysis.

In the absence of this proof, the proposal that a cytoplasmic glycosylation machinery may exist seems less likely than other possibilities that could be discussed. For instance, the authors do offer the possibility that NGLY1 has functions independent of its enzymatic activity. Perhaps one of those activities influences the efficiency of the oligosaccharyltransferase complex such that it is less likely to modify both of the extracellular Asn residues. Loss of NGLY1 might then lead to more efficient glycosylation of NKCC1. This mechanism does not require the invocation of an unknown cytoplasmic N-linked glycosylation machinery. A more well-reasoned discussion of other possibilities is warranted.

This is a good suggestion and we have added the following sentences to the Discussion:

“If this effect is secondary, it may be that NGLY1 is directly affecting an oligosaccharyltransferase complex that in turn modifies the already present glycans. This hypothesis is supported by the fact that Endo H treatment did not affect NKCC1 from either +/+ or -/- cells, indicating that NKCC1 has been fully processed through the Golgi.”.

Reviewer #3:[…]1) Although most of this work is well done, the data does not support the conclusion that NKCC1 is a direct substrate for NGLY1. As the authors point out, the two predicted N-glycans are on an extracelluar loop of NKCC1 and are of the complex-type, so it is unlikely that a cytoplasmically localized NGLY1 could access the sites to de-glycosylate them. Instead, the authors invoked a cytoplasmic N-glycosylation event with an unknown cytoplasmic glycosyltransferase modifying N168, which is cytoplasmic. The authors reference a review by Hart and Wells, 2017, which describes a report that the α-subunit of the dog kidney sodium pump Na+, K^+^-ATPase is modified on a cytoplasmic domain with an N-glycan. The paper referred to was published in 1990 (https://www.ncbi.nlm.nih.gov/pubmed/2175915 ) and has not been rigorously confirmed using mutagenesis or mass spectral glycoproteomic site mapping. In the review, Hart and Wells conclude, "This provocative claim has remained unresolved". In addition, the large shift in size in Figure 4A, estimated to be as much as 20 kDa (Figure 4C) suggests that the change is much more than a simple complex-type N-glycan. A typical biantennary complex N-glycan with two sialic acids has a mass of just over 2 kDa. This suggests that the N-glycans on the extracellular loop of NKCC1 are extended in some way, possibly by poly-N-acetyl-lactosamine repeats or polysialic acid. Thus, a much simpler explanation for the change in molecular weight in Figure 4A is that loss of NGLY1 in these cells induces, indirectly, expression of glycosyltransferases responsible for enhanced extension of the 2 N-glycans on the extracellular loop of NKCC1. Without more experimentation, the authors cannot conclude that NKCC1 is a direct substrate of NGLY1. Other possibilities are more likely.

We agree that this conclusion was unwarranted by the data and have changed all mentions of it in the manuscript. In regards to the hypothesis about glycosyltransferases we have added these sentences to the Discussion: “If this effect is secondary, it may be that NGLY1 is directly affecting an oligosaccharyltransferase complex that in turn modifies the already present glycans. This hypothesis is supported by the fact that Endo H treatment did not affect NKCC1 from either +/+ or -/- cells, indicating that NKCC1 has been fully processed through the Golgi.”.

2) It is not clear why KD of Ncc69 rescued the Pngl phenotype in the initial screen while KD of both in Figure 3A caused synthetic lethality.

The initial screen did not utilize KD of any genes aside from Pngl, but rather used natural genetic variation in the DGRP fly lines. None of the SNPs associated with Ncc69 are known to cause a decrease in the expression levels. We have tried to make this more clear by adding a plot of the Ncc69 expression from each DGRP line versus our survival data (Figure 1—figure supplement 1), and adding language into the Results section.

[Editors’ note: what follows is the authors’ response to the second round of review.]

Please address the following points:Impact statement: Please add "potentially" before "explaining several symptoms of NGLY1 deficiency such as lack of sweat and tears.".

We have added the word “potentially”.

The authors write "there is no evidence of ER stress upon RNAi knockdown of NGLY1 in *Drosophila*, (Owings et al., 2018) or upon loss of NGLY1 function in mouse, rat, and human cells (Asahina et al., 2020; Mueller et al., 2020; Tambe, Ng and Freeze, 2019)." Also, in the Discussion a similar statement has been made. Loss of Ngly1 in mouse embryonic fibroblasts has recently been shown to significantly increase the level of several ER stress markers (BiP, phopho-IRE1α and OS9), and this was further enhanced when an NGLY1 substrate was overexpressed in these cells (Galeone et al., 2020). This paper has been already cited by the authors in a different part of the manuscript. Please revise these sentence to incorporate the findings of this report.

The text have been changed as such “While a recent report showed that ER stress markers were increased in NGLY1 -/- MEFs (Galeone et al., 2020), other experiments such as RNAi knockdown of NGLY1 in *Drosophila*, (Owings et al., 2018) and loss of NGLY1 function in mouse, rat, and human cells (Asahina et al., 2020; Mueller et al., 2020; Tambe, Ng and Freeze, 2019) have shown no evidence of ER stress.”

And “While perturbations to ERAD often result in ER stress, we have previously reported that there was no functional or transcriptome evidence for ER stress in a *Drosophila* model of NGLY1 deficiency (Owings et al., 2018). Others have reported no ER stress in NGLY1 -/- human cells, mice, and rats (Asahina et al., 2020; Mueller et al., 2020; Tambe et al., 2019). However, there is conflicting evidence for ER stress as it was recently reported that ER stress markers were upregulated in NGLY -/- MEFs (Galeone et al., 2020). Nevertheless, in our current screen, we did not identify any genes involved in canonical ER stress responses, suggesting that ER stress might not play a large role in the pathogenesis of the disease.”

In the response letter, the authors have acknowledged that the PNGase F and Endo H data are difficult to interpret and have agreed with the reviewers' comments that the data are not sufficient for calling NKCC1 a target of NGLY1. In accordance with this, the following sentence (between quotation marks) should be removed, as it implies otherwise (The authors wrote: In cells derived from an NGLY1 -/- mouse model we found that NKCC1 protein migrated at a higher molecular weight relative to +/+ cells. "Treatment with PNGase eliminated this size difference, confirming that this is due to N-linked glycosylation.")

We have removed this sentence.

A similar sentence that needs to be removed, as it is not supported by the data: "but rather NKCC1 has a molecular weight shift that can be eliminated with PNGase treatment".

This sentence has been changed to read: “Given that we show the abundance of NKCC1 does not change, but rather NKCC1 has a molecular weight shift, it is likely that the altered state is due to some difference in a post-translational modification.”.